# Interplay of BAF and MLL4 promotes cell type-specific enhancer activation

Young-Kwon Park[1,5], Ji-Eun Lee [1,5], Zhijiang Yan[2,3], Kaitlin McKernan [1], Tommy O'Haren[1], Weidong Wang [2], Weiqun Peng[4] & Kai Ge [1✉]

Cell type-specific enhancers are activated by coordinated actions of lineage-determining transcription factors (LDTFs) and chromatin regulators. The SWI/SNF chromatin remodeling complex BAF and the histone H3K4 methyltransferase MLL4 (KMT2D) are both implicated in enhancer activation. However, the interplay between BAF and MLL4 in enhancer activation remains unclear. Using adipogenesis as a model system, we identify BAF as the major SWI/SNF complex that colocalizes with MLL4 and LDTFs on active enhancers and is required for cell differentiation. In contrast, the promoter enriched SWI/SNF complex PBAF is dispensable for adipogenesis. By depleting BAF subunits SMARCA4 (BRG1) and SMARCB1 (SNF5) as well as MLL4 in cells, we show that BAF and MLL4 reciprocally regulate each other's binding on active enhancers before and during adipogenesis. By focusing on enhancer activation by the adipogenic pioneer transcription factor C/EBPβ without inducing cell differentiation, we provide direct evidence for an interdependent relationship between BAF and MLL4 in activating cell type-specific enhancers. Together, these findings reveal a positive feedback between BAF and MLL4 in promoting LDTF-dependent activation of cell type-specific enhancers.

[1] Adipocyte Biology and Gene Regulation Section, National Institute of Diabetes and Digestive and Kidney Diseases, National Institutes of Health (NIH), Bethesda, MD, USA. [2] Laboratory of Genetics and Genomics, National Institute on Aging, NIH, Baltimore, MD, USA. [3] School of Basic Medical Sciences, Wenzhou Medical University, Wenzhou, China. [4] Department of Physics, The George Washington University, Washington, DC, USA. [5]These authors contributed equally: Young-Kwon Park, Ji-Eun Lee. ✉email: kai.ge@nih.gov

Throughout development, chromatin architecture undergoes dynamic changes that are critical for promoting appropriate cell type-specific enhancer activation and gene expression. These changes are coordinated by the action of transcription factors (TFs) and epigenomic regulators, including ATP-dependent chromatin remodeling complexes, which modulate chromatin accessibility and gene expression through the mobilization of nucleosomes[1,2]. The large multi-subunit SWItch/Sucrose Non-Fermentable (SWI/SNF) chromatin remodeling complexes have been shown to play a crucial role in cellular and tissue development[1]. SWI/SNF mutations are also frequently associated with human diseases, including over 20% of cancers and a variety of neurodevelopmental disorders[3,4].

Mammalian SWI/SNF complexes exist in three different isoforms with distinct subunit compositions: BRG1/BRM-associated factor (BAF), polybromo-associated BAF (PBAF), and noncanonical GLTSCR1L-containing BAF (GBAF, also known as ncBAF)[5,6]. All three complexes contain an ATPase catalytic subunit, either SMARCA4 (BRG1) or SMARCA2 (BRM), and an initial core, composed of SMARCC1 (BAF155), SMARCC2 (BAF170), and SMARCD1/D2/D3 (BAF60A/B/C)[5,7]. The BAF complex contains SMARCB1 (SNF5, INI1, or BAF47), SMARCE1 (BAF57), and SS18 and BAF-specific subunits ARID1A/B (BAF250A/B) and DPF1/2/3 (Fig. 1a). Like BAF, PBAF complex also consists of the initial core, SMARCB1 and SMARCE1 as well as PBAF-specific subunits PHF10, ARID2 (BAF200), BRD7, and PBRM1. The recently characterized GBAF is a smaller complex containing SMARCA4 or SMARCA2, the initial core, SS18, and GBAF-specific subunits GLTSCR1/L and BRD9[5–7]. These three distinct complexes exhibit different genomic localization patterns. BAF primarily binds on enhancers[8–10]. Conversely, PBAF is more localized on promoter regions, and GBAF binding sites are enriched with the CTCF motif[11,12].

Enhancers are cis-acting regulatory regions containing TF binding sites and are responsible for facilitating cell type-specific gene expression through communication with promoters[13]. Enhancers are marked by H3K4 mono-methylation (H3K4me1), which is primarily placed by the partially redundant H3K4 methyltransferases MLL3 (KMT2C) and MLL4 (KMT2D)[14,15]. Along with H3K4me1, active enhancers are further marked with histone acetyltransferases CBP/p300-catalyzed H3K27 acetylation (H3K27ac)[16,17]. MLL3 and MLL4 are required for enhancer activation and cell type-specific gene expression during cell differentiation[14]. Specifically, MLL3/MLL4 controls enhancer activation by facilitating CBP/p300 binding on enhancers[18]. MLL3 and MLL4 are essential for embryonic development as well as the development of multiple tissues, including adipose and muscle[14,19,20].

Of the three SWI/SNF complexes, BAF primarily interacts with enhancer regions. Recent studies have demonstrated that BAF is required for enhancer activation[8–10]. Following the deletion of BAF catalytic subunit SMARCA4 and core subunit SMARCB1, H3K27ac levels on enhancers are decreased, and enhancers fail to be activated. The deletion of SMARCB1 results in the destabilization of the BAF complex on the genome. BAF-specific subunit ARID1A also plays a major role in BAF-mediated enhancer activation. Particularly, the loss of ARID1A has been shown to diminish BAF complex occupancy on enhancers[21]. SMARCA4 interacts with LDTFs including the adipogenic pioneer factor C/EBPβ and is required for adipogenesis[22–25]. Although the BAF complex plays an essential role in enhancer activation and development, many questions remain regarding how it interacts with other epigenomic regulators during enhancer activation.

In this study, we evaluated the interrelationship between BAF and MLL4 on cell type-specific enhancers using adipogenesis and the pioneer factor C/EBPβ-mediated enhancer activation as

model systems. Loss of SWI/SNF subunits in cells and mice provides evidence that BAF, but not PBAF, is critical for enhancer activation, adipogenesis and adipocyte gene expression. Given that MLL4 is similarly important for enhancer activation, cell differentiation and cell type-specific gene expression, we investigated the relationship between BAF and MLL4 before and during adipogenesis, revealing that BAF and MLL4 colocalize with LDTFs on active enhancers and facilitate enhancer activation in an interdependent manner. We further uncover the reciprocal regulation between BAF and MLL4 in adipogenic pioneer TF C/EBPβ-mediated enhancer activation.

## Results

**Expression and genomic binding of SWI/SNF subunits in adipogenesis.** To explore the role of each SWI/SNF complex in cell differentiation and cell type-specific enhancer activation, we used differentiation of preadipocytes towards adipocytes (adipogenesis) as a model system. We first explored the expression profiles of SWI/SNF subunits at four key time points in adipogenesis: actively growing preadipocytes (day −3, D-3), overconfluent preadipocytes before the induction of differentiation (day 0, D0), immature adipocytes during differentiation (day 2, D2), and adipocytes after differentiation (day 7, D7)[18]. By RNA-Seq analysis, we found that all SWI/SNF subunits except *Dpf3*, a BAF-specific subunit, were expressed in adipogenesis. The expression levels of SWI/SNF subunits were largely invariant from D-3 to D7. Among the two enzymatic subunits, *Smarca4* was expressed at significantly higher levels than *Smarca2* throughout the differentiation (Fig. 1b and Supplementary Fig. 1a).

Next, we performed ChIP-Seq analyses to profile genomic localizations of BAF, PBAF, and GBAF complexes before (D-3) and during (D2) adipogenesis. We chose the enzymatic subunit SMARCA4, SS18, BAF-specific subunit ARID1A, PBAF-specific subunit ARID2 and GBAF-specific subunit BRD9. Each subunit exhibited a differentiation stage-specific genomic binding (Supplementary Fig. 1b). ChIP-Seq identified distinct genomic distribution of each subunit in four types of regulatory elements: active enhancer (AE), primed enhancer, promoter and other regions defined as in our previous report[14]. SMARCA4 binding sites were distributed in all types of regulatory elements, but primarily on primed enhancers and AEs. While SS18 and ARID1A binding sites were mainly located on AEs, PBAF-specific ARID2 was strongly enriched on promoter regions especially during adipogenesis. GBAF-specific BRD9 binding sites were enriched on AEs and other regions (Fig. 1c and Supplementary Fig. 1c). By motif analysis of the top 3,000 binding sites of each subunit, we found that SMARCA4, SS18 and ARID1A binding regions were enriched with motifs of AP-1 family TFs Jdp2, JunD and Jun in preadipocytes (D-3), but with motifs of adipogenic TFs such as C/EBPα, C/EBPβ, and ATF4 during adipogenesis (D2). ARID2 and BRD9 binding regions were selectively enriched with motifs of ETS family TFs and CTCF, respectively (Fig. 1d and Supplementary Fig. 1d). These results demonstrate the distinct targeting of each complex in adipogenesis: BAF to enhancers, PBAF to promoters and GBAF to CTCF sites.

Further, we defined confident genomic binding regions of BAF (16,003), PBAF (2,063) and GBAF (1,053) at D2 of adipogenesis by selecting overlapping binding sites between SMARCA4 and complex-specific subunits (Fig. 1e and Supplementary Fig. 1e). Even though SS18 was reported as a common subunit in BAF and GBAF complexes, the majority of SS18 genomic binding sites (20,257/22,118) overlapped with those of BAF-specific ARID1A, and only a small fraction (791/22,118) overlapped with those of

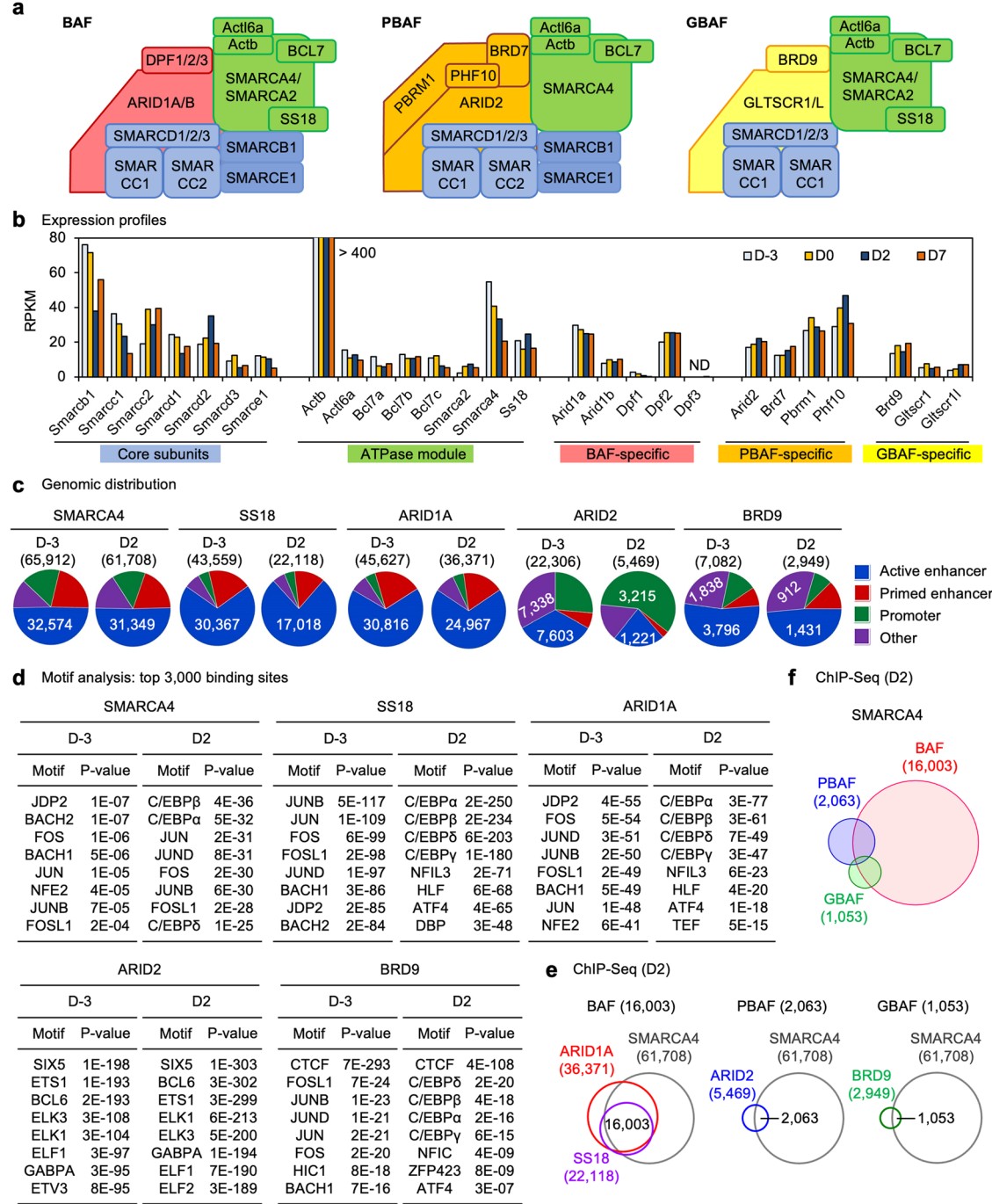

**Fig. 1 Expression and genomic binding of SWI/SNF subunits in adipogenesis. a** Schematic of subunits in SWI/SNF complexes BAF, PBAF and GBAF (ncBAF). **b** Expression profiles of SWI/SNF subunits in adipogenesis of immortalized brown preadipocytes ($n = 1$). Published RNA-Seq data sets were used (GSE74189)[18]. D-3, day -3. D2, day 2. Expression profiles of SWI/SNF subunits in adipogenesis of 3T3-L1 white preadipocytes and a different brown preadipocyte cell line are presented in Supplementary Fig. 1a. **c** Genomic distributions of SMARCA4 (BRG1), SS18, BAF-specific ARID1A, PBAF-specific ARID2 and GBAF-specific BRD9 before (D-3) and during (D2) adipogenesis of brown preadipocytes were determined by ChIP-Seq. Promoters were defined as transcription start sites ±1 kb. Active enhancers were defined as H3K4me1+ H3K27ac+ promoter-distal regions. Primed enhancers were defined as H3K4me1+ H3K27ac− promoter-distal regions. Numbers of binding sites are indicated. **d** Motif analysis of SWI/SNF subunits binding regions at D-3 and D2 of adipogenesis using SeqPos motif tool. Top 3,000 significant binding regions were used. **e** Venn diagrams depicting genomic binding of SMARCA4, SS18, ARID1A, ARID2 and BRD9 at D2 of adipogenesis. BAF, PBAF, and GBAF binding sites were defined as SMARCA4+ SS18+ ARID1A+, SMARCA4+ ARID2+, and SMARCA4+ BRD9+ regions, respectively. **f** Venn diagram depicting genomic binding of BAF, PBAF and GBAF at D2 of adipogenesis.

GBAF-specific BRD9. Among 17,165 SWI/SNF-associated SMARCA4 binding regions, BAF (16,003) exhibited substantially greater genomic occupancy than PBAF (2,063) and GBAF (1,053) during adipogenesis (Supplementary Fig. 1e). Only a small subset (1,504/16,003, 9.4%) of BAF binding sites overlapped with those of PBAF or GBAF (Fig. 1f). These findings suggest that among the three SWI/SNF complexes, BAF is the major regulator of enhancers in adipogenesis.

**BAF, but not PBAF, is required for adipogenesis**. To address the functions of SWI/SNF complexes in adipogenesis, we first depleted endogenous SMARCA4, the catalytic ATPase subunit, using the auxin-inducible degron (AID) system[26]. Primary brown preadipocytes were isolated from newborn pups of Smarca4[AID/AID] mice, which were generated using CRISPR-mediated insertion of an AID tag in the C-terminus of endogenous Smarca4 alleles. Cells were immortalized and infected with a retroviral vector expressing the Myc-tagged auxin receptor Tir1. In the presence of auxin, Tir1 binds to the AID tag and induces proteasome-dependent degradation of endogenous SMARCA4 (Fig. 2a). As shown in Fig. 2b, auxin induced a rapid Tir1-dependent depletion of endogenous SMARCA4 protein within 4 h. Depletion of SMARCA4 prevented the appearance of lipid droplets, indicating that SMARCA4 is essential for adipogenesis in cell culture (Fig. 2c).

Next, we examined the role of SMARCB1 in adipogenesis. We generated a conditional knockout (cKO) of SMARCB1 in preadipocytes by crossing Smarcb1[f/f] mice with PdgfRα-Cre mice. The PdgfRα-Cre transgene is expressed in most adipocyte precursor cells[27]. Smarcb1 cKO mice (Smarcb1[f/f];PdgfRα-Cre) died shortly after birth, presumably due to severe defects in cranial development (Supplementary Fig. 2a-d). Immunohistochemical analysis revealed that deletion of Smarcb1 led to a severe reduction of interscapular brown adipose tissue (BAT) in E18.5 embryos (Fig. 2d and Supplementary Fig. 2e). Consistent with this phenotype, deletion of Smarcb1 reduced the expression of adipogenesis markers Pparg, Cebpa, and Fabp4 as well as BAT markers Prdm16 and Ucp1, but not myogenesis markers (Supplementary Fig. 2f). These findings indicate that SMARCB1 is required for adipose tissue development in mice.

To understand how SMARCB1 regulates adipogenesis, we isolated and immortalized Smarcb1[f/f] brown preadipocytes. Deletion of Smarcb1 in immortalized brown preadipocytes did not affect growth rates (Fig. 2e). Consistent with previous data[9], deletion of Smarcb1 did not affect the expression of SMARCA4 nor the interaction between SMARCA4 and core subunit SMARCC2 (Fig. 2f and Supplementary Fig. 2g). However, deletion of Smarcb1 blocked adipogenesis (Fig. 2g). We performed RNA-Seq analysis before (D-3) and during (D2) adipogenesis. Using a 2.5-fold cut-off for differential expression, we defined 937 and 1,382 genes that were up- and down-regulated from D-3 to D2 of differentiation (Supplementary Fig. 2h). Among the 937 up-regulated genes, 362 were induced in a SMARCB1-dependent manner. GO analysis showed that these genes were strongly associated functionally with fat cell differentiation and lipid metabolism (Supplementary Fig. 2i). Deletion of Smarcb1 did not affect the induction of pioneer TF C/EBPβ or the expression of other SWI/SNF subunits, but blocked the induction of adipocyte marker genes Pparg, Cebpa, and Fabp4 (Fig. 2f, h and Supplementary Fig. 2j), suggesting that SMARCB1 is required for the induction of adipocyte genes downstream of C/EBPβ.

Because SMARCA4 and SMARCB1 are common subunits in BAF and PBAF, we wanted to clarify the function of each complex in adipogenesis. To determine the role of BAF in adipogenesis, we infected preadipocytes with lentiviral shRNA targeting BAF-specific subunit ARID1A. Knockdown of ARID1A did not affect SMARCA4 and SMARCB1 expression or pioneer TF Cebpb expression, but it severely impaired induction of Pparg and Fabp4 and adipogenesis (Fig. 2i–k). To study the role of PBAF in adipogenesis, we generated a cKO of PBAF-specific subunit PBRM1 in precursor cells of brown adipocytes by crossing Pbrm1[f/f] mice with Myf5-Cre mice. Myf5-Cre is specifically expressed in somitic precursor cells for both BAT and skeletal muscle in the interscapular region of mice[14].

Pbrm1[f/f];Myf5-Cre (cKO) pups were born at the expected Mendelian ratio and survived to adulthood without obvious developmental defects (Supplementary Fig. 3a, b). The adipose tissues isolated from cKO mice were similar in size and morphology to the control mice (Supplementary Fig. 3c, d). Although the Pbrm1 gene level was reduced by over 80% in the BAT of cKO mice, deletion of Pbrm1 did not significantly affect the expression of adipogenesis markers Pparg, Cebpa, and Fabp4 or BAT marker Ucp1 (Supplementary Fig. 3e). To further understand the role of PBRM1 in adipogenesis, we induced differentiation of primary brown preadipocytes isolated from cKO mice. Deletion of Pbrm1 by Myf5-Cre had little effects on adipogenesis in culture (Supplementary Fig. 3f, g). By RNA-Seq analysis at D7 of adipogenesis, we found that only a small number of genes were increased (0.4%) or decreased (0.4%) over 2.5-fold in Pbrm1 cKO cells compared with control cells (Supplementary Fig. 3h). Differentially regulated genes were not associated with adipocyte differentiation or lipid metabolism (Supplementary Fig. 3i, j). Taken together, these findings suggest that BAF, but not PBAF, is essential for adipogenesis.

**BAF co-localizes with LDTFs and MLL4 on cell type-specific enhancers during adipogenesis**. To assess the genomic binding of BAF, we first selected SMARCA4+ AEs (31,349) or promoters (8,871), then examined changes in genomic binding of BAF subunits SS18 and ARID1A before (D-3) and during (D2) adipogenesis. While genomic binding of the common ATPase subunit SMARCA4 was found on both AEs and promoters, genomic bindings of SS18 and ARID1A were mainly localized on AEs and increased from D-3 to D2 (Fig. 3a). Motif analysis of the top 3,000 BAF-bound (SMARCA4+ SS18+) AEs showed enrichment of AP-1 family TF motifs at D-3 and motifs of adipogenic TFs C/EBPs and ATF4 at D2 (Fig. 3b and Supplementary Fig. 4a–c). These results indicate that BAF is enriched on cell type-specific enhancers during adipogenesis.

To obtain insights into how LDTFs coordinate with BAF on AEs, we analyzed the genomic localization of adipogenic TFs C/EBPβ, C/EBPα and PPARγ, which are sequentially induced to activate adipocyte-specific genes in adipogenesis[28,29]. Among the 10,813 BAF-bound AEs during adipogenesis (D2), BAF was already bound to 4,410 of them in preadipocytes (D-3) and maintained the binding at D2 (BAF-prebound AEs). In contrast, 6,403 AEs exhibited emergent BAF binding at D2 but not at D-3 (BAF-de novo AEs) (Fig. 3c). Consistent with motif analysis, BAF was highly co-localized with C/EBPβ, C/EBPα and PPARγ on AEs at D2 of adipogenesis. Particularly, C/EBPβ binding was detected on BAF-prebound AEs at D-3 but increased markedly from D-3 to D2 on BAF-de novo AEs (Fig. 3d, e). Since MLL4 and CBP also colocalize with LDTFs on cell type-specific AEs during adipogenesis[18], we assessed the co-occupancy among BAF, MLL4 and CBP. We found that MLL4 and CBP co-localized with BAF on both BAF-prebound and BAF-de novo AEs. While MLL4 and CBP binding on BAF-prebound AEs remained largely unchanged from D-3 to D2 of adipogenesis, their binding on BAF-de novo AEs was induced. Accordingly, the levels of MLL3/MLL4−catalyzed H3K4me1 and CBP/p300-catalyzed H3K27ac also increased on BAF-de novo AEs from D-3 to D2 (Fig. 3d, e). Colocalization of BAF with LDTFs, MLL4 and CBP on BAF-de novo AEs was also found on Pparg and Cebpa gene loci during adipogenesis (Supplementary Fig. 4d, e). ATAC-Seq analysis revealed that chromatin accessibility was highly correlated with BAF binding intensities on AEs during adipogenesis. The induction of chromatin accessibility on BAF-de novo AEs coincided with the increase in the binding of LDTFs, MLL4 and CBP on AEs (Fig. 3d, e and Supplementary Fig. 4d, e). These

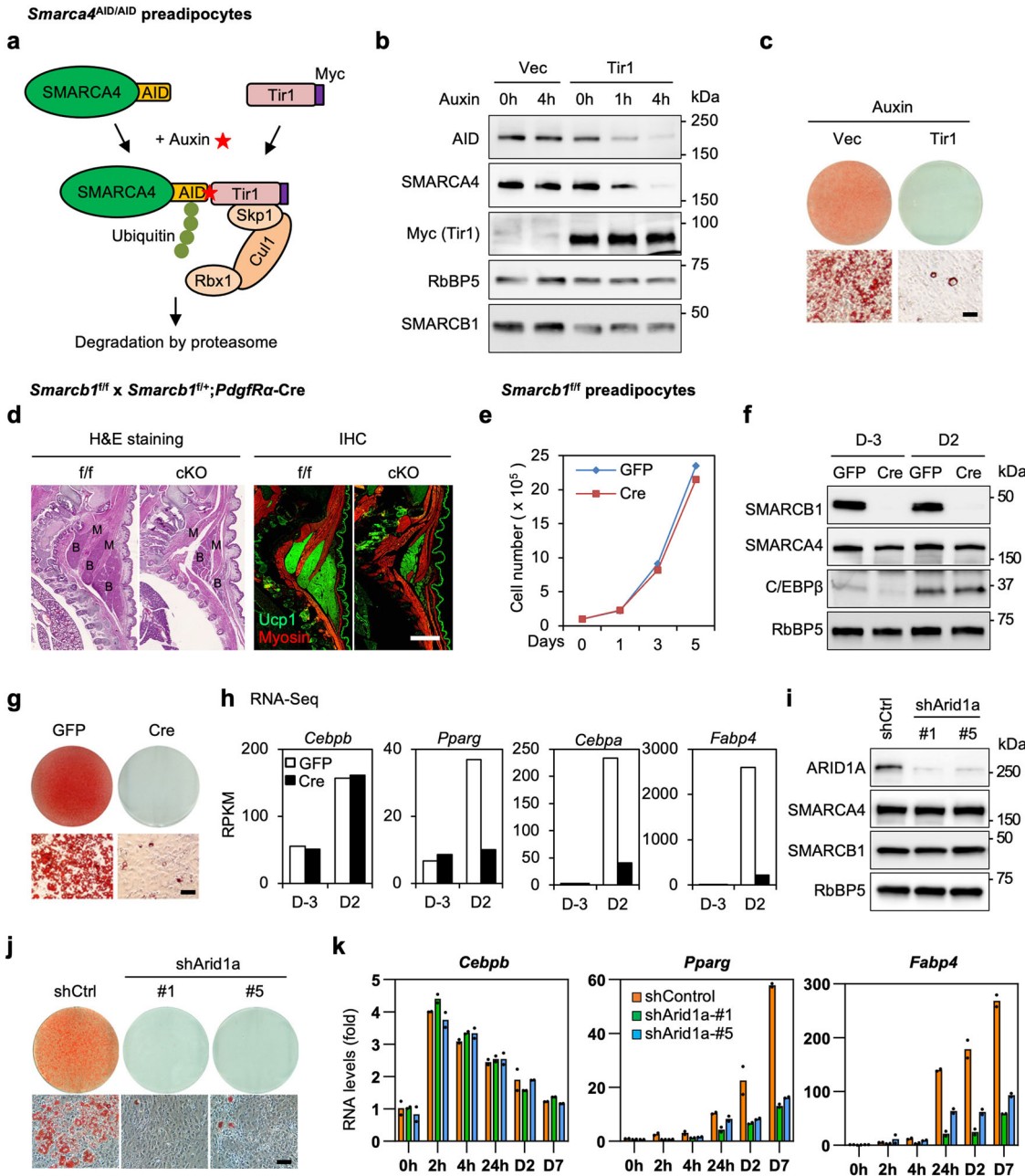

**Fig. 2 BAF subunits SMARCA4, SMARCB1 and ARID1A are required for adipogenesis. a–c** Depletion of SMARCA4 by auxin-inducible degron (AID) system inhibits adipogenesis. SV40T-immortalized *Smarca4*^AID/AID brown preadipocytes were infected with retroviruses expressing Tir1 or vector (Vec), and then treated with auxin for indicated times. **a** A schematic illustration of the auxin-induced depletion of endogenous SMARCA4 protein in Tir1-expressing *Smarca4*^AID/AID cells. TIR1, transport inhibitor response 1; Cul1, Cullin-1; Rbx1, RING-box protein 1; Skp1, S-phase kinase-associated protein 1. **b** Western blot analyses using antibodies indicated on the left. RbBP5 was used as a loading control. $n = 3$ biological replicates. **c** Oil Red O staining at day 7 (D7) of adipogenesis. Cells were treated with auxin throughout the differentiation. $n = 2$ biological replicates. Scale bar = 50 μm. **d** SMARCB1 is essential for brown adipogenesis in vivo. Sections of the interscapular area of E18.5 Smarcb1^f/f (f/f) and *Smarcb1*^f/f;*PdgfRα-Cre* (cKO) embryos were stained with H&E (left panels) or with antibodies recognizing brown adipose tissue (BAT) marker Ucp1 (green) and skeletal muscle marker Myosin (red) (right panels). B, BAT; M, muscle. Scale bar = 1 mm. $n = 2$ biological replicates. **e–h** SMARCB1 is essential for adipogenesis in culture. SV40T-immortalized *Smarcb1*^f/f brown preadipocytes were infected with adenoviral GFP or Cre, followed by adipogenesis assays. **e** Deletion of *Smarcb1* does not affect cell growth rates of immortalized brown preadipocytes. $1 \times 10^5$ preadipocytes were plated and the cumulative cell numbers were determined for 5 days ($n = 2$). Data are presented as mean values ± std. dev. **f** Before (D-3) and during (D2) adipogenesis, nuclear extracts were analyzed by Western blot using indicated antibodies. $n = 2$ biological replicates. **g** Oil Red O staining at D7 of adipogenesis. $n = 3$ biological replicates. Scale bar = 50 μm. **h** Expression of *Cebpb*, *Pparg*, *Cebpa* and *Fabp4* before (D-3) and during (D2) adipogenesis was determined using RNA-Seq ($n = 1$). RPKM values indicate gene expression levels. **i–k** ARID1A is required for adipogenesis in culture. Immortalized *Smarcb1*^f/f brown preadipocytes were infected with lentiviral vectors expressing control (shCtrl) or *Arid1a* knockdown shRNA (shArid1a), followed by adipogenesis assays. **i** Western blot analysis of ARID1A in preadipocytes. $n = 2$ biological replicates. **j** Oil Red O staining at D7 of adipogenesis. $n = 3$ biological replicates. Scale bar = 50 μm. **k** qRT-PCR analysis of *Cebpb*, *Pparg*, and *Fabp4* expression at indicated time points of adipogenesis. Expression levels were normalized to *18 S rRNA*. $n = 2$ biologically independent samples. Data are presented as mean values ± std. dev.

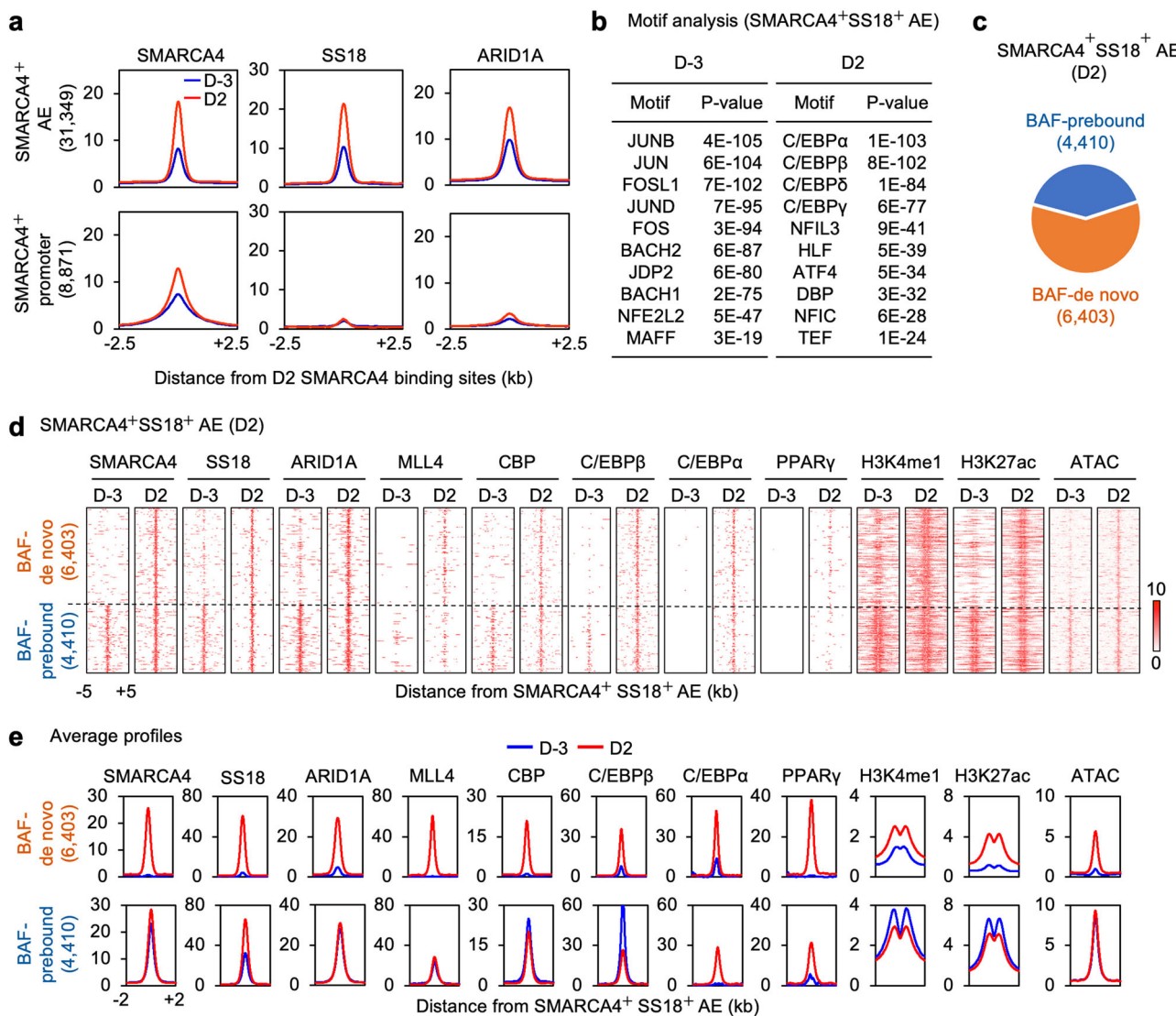

**Fig. 3 BAF co-localizes with LDTFs and MLL4 on active enhancers during adipogenesis. a** Average binding profiles of SMARCA4, SS18, and ARID1A around the center of SMARCA4+ active enhancers (AEs) or promoters before (D-3) and during (D2) adipogenesis of brown preadipocytes. Normalized read counts are shown. **b** Motif analysis of the top 3,000 SMARCA4+ SS18+ AEs using SeqPos motif tool. **c** Pie chart depicting BAF-binding (SMARCA4+ SS18+) AEs at D2 of adipogenesis. Among the 10,813 BAF-binding AEs at D2, 6,403 are de novo BAF-binding AEs (BAF-de novo), while 4,410 AEs are already bound by BAF (BAF-prebound) at D-3. **d**, **e** BAF subunits SMARCA4, SS18 and ARID1A colocalize with LDTFs (C/EBPβ, C/EBPα, and PPARγ), MLL4, CBP and open chromatin on AEs (H3K4me1+ H3K27ac+) at D2 of adipogenesis. The local epigenetic environment exhibits coordinated changes during differentiation. Heat maps (**d**) and average profiles (**e**) were aligned around the center of SMARCA4+ AEs. Published ChIP-Seq data sets for MLL4, CBP, C/EBPβ, C/EBPα, and PPARγ were used (GSE74189)[18].

results indicate that BAF colocalizes with LDTFs, MLL4 and CBP on cell type-specific enhancers and promotes chromatin accessibility during differentiation.

At D2 of adipogenesis, about 40% (4,362/10,813) of BAF-bound AEs were occupied by MLL4 (Supplementary Fig. 5a). Differential motif analysis of BAF-bound AEs revealed that motifs of LDTFs PPARγ and C/EBPs were enriched on MLL4+ AEs while motifs of the AP-1 family TFs were enriched on MLL4− AEs at D2 (Supplementary Fig. 5b). Heat maps and average profiles also showed that from D-3 to D2 of adipogenesis, the binding of LDTFs and BAF subunits (SMARCA4, SS18 and ARID1A) was more significantly increased on MLL4+ than MLL4− AEs. Particularly, levels of CBP and H3K27ac were only induced on MLL4+ AEs (Supplementary Fig. 5c, d). D2 MLL4− AEs were already accessible before adipogenesis (D-3). In contrast, chromatin accessibility on D2 MLL4+ AEs was dramatically increased

from D-3 to D2 (Supplementary Fig. 5c, d). These results suggest the involvement of MLL4 in enhancer activation by BAF.

To test the possibility of physical interaction between BAF and MLL4, we identified proteins associated with endogenous MLL4 and UTX, a subunit of the MLL4 complex, in mouse embryonic stem cells (ESCs) by mass spectrometry. Consistent with previous results[30], while immunoprecipitation with an anti-UTX antibody pulled down both MLL3 and MLL4 as well as other components of MLL3/MLL4 complexes, MLL3 was not co-immunoprecipitated by an anti-MLL4 antibody. Notably, several SWI/SNF subunits, including SMARCA4, SMARCC1, SMARCD1, SMARCB1 and SMARCE1, were also pulled down by anti-MLL4 and/or anti-UTX antibody from ESC nuclear extracts (Fig. 4a and Supplementary Data 1 and 2). We also confirmed that endogenous SMARCA4 and SMARCB1 are associated with endogenous MLL4 and UTX, but not the MLL1 complex subunit Menin, in HEK293T cells and

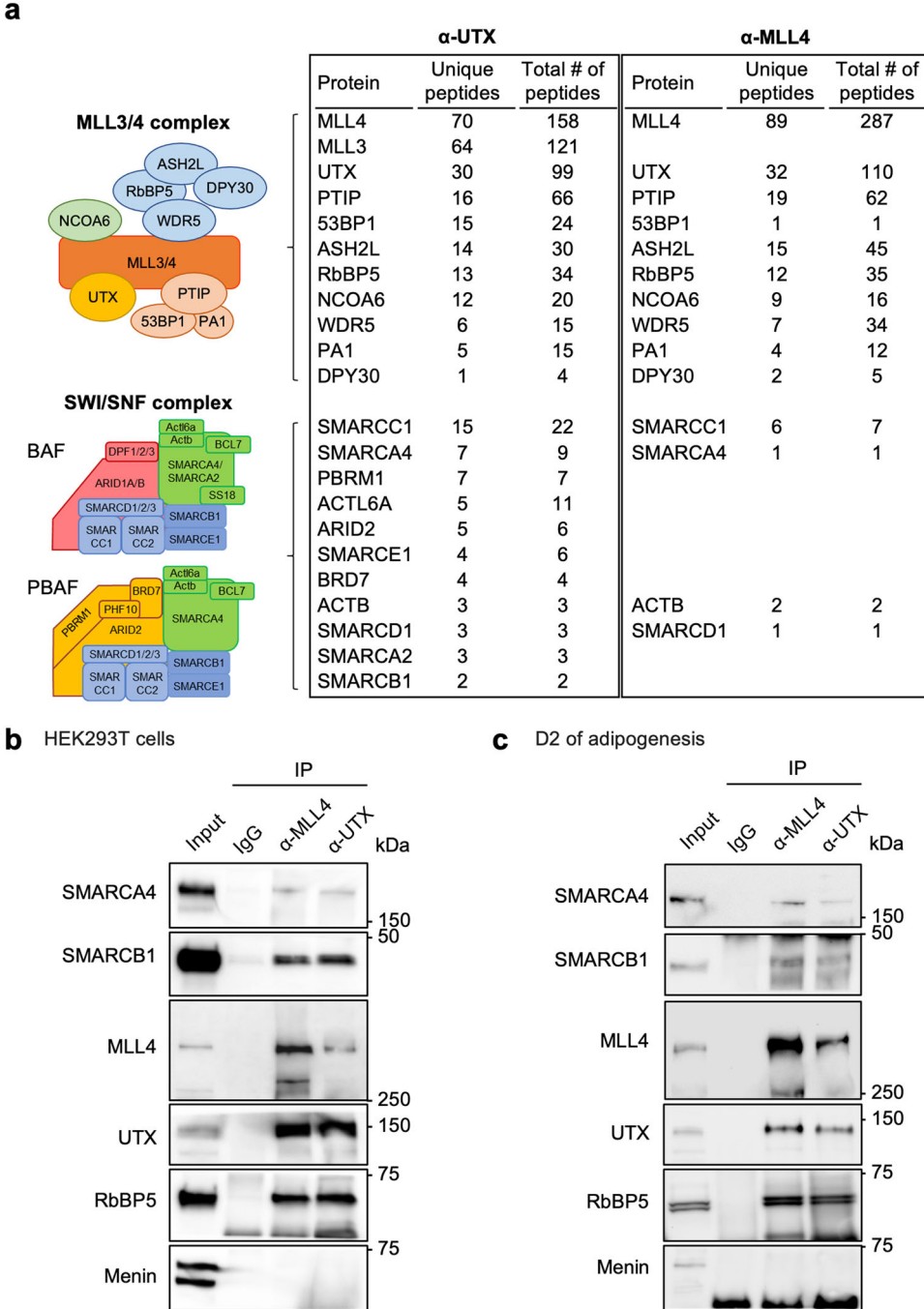

**Fig. 4 UTX and MLL4 associate with SWI/SNF complex subunits in cells. a** UTX and MLL4 associate with SWI/SNF components in mouse embryonic stem cells. Nuclear extracts were subjected to immunoprecipitation with anti-UTX or anti-MLL4 antibodies. Components of MLL3/4 and SWI/SNF complexes identified by mass spectrometry after immunoprecipitation of endogenous UTX or MLL4 are presented ($n = 1$). Unique and total peptide numbers for IP-enriched BAF and PBAF complex subunits as well as MLL4 complex subunits are shown. Full lists of identified proteins are presented in the Supplementary Data 1 and 2. **b**, **c** MLL4 complex associates with SMARCA4 and SMARCB1 in cells. Nuclear extracts from HEK293T cells (**b**) or preadipocytes during (D2) adipogenesis (**c**) were subjected to immunoprecipitation with anti-MLL4 or anti-UTX antibody. Immunoprecipitates were analyzed by Western blotting with antibodies indicated on the left. $n = 2$ biological replicates.

preadipocytes at D2 of adipogenesis (Fig. 4b, c). Together, these findings suggest that BAF cooperates with MLL4 to activate cell type-specific enhancers during differentiation.

**Reciprocal regulation between BAF and MLL4 on active enhancers during adipogenesis.** To study the relationship between BAF and MLL4 on cell type-specific enhancers, we

assessed their genomic binding on adipogenic enhancers at D2 of differentiation. Adipogenic enhancers were defined as AEs that are bound by C/EBPs or PPARγ (Fig. 5a). This set of enhancers are mostly de novo. We first asked whether MLL4 is required for BAF recruitment to adipogenic enhancers during differentiation. To eliminate the compensatory effect of MLL3, we deleted *Mll4* in *Mll3*$^{−/−}$*Mll4*$^{f/f}$ cells[14]. Consistent with our previous data, deletion of *Mll4* completely blocked adipogenesis (Supplementary Fig. 6a).

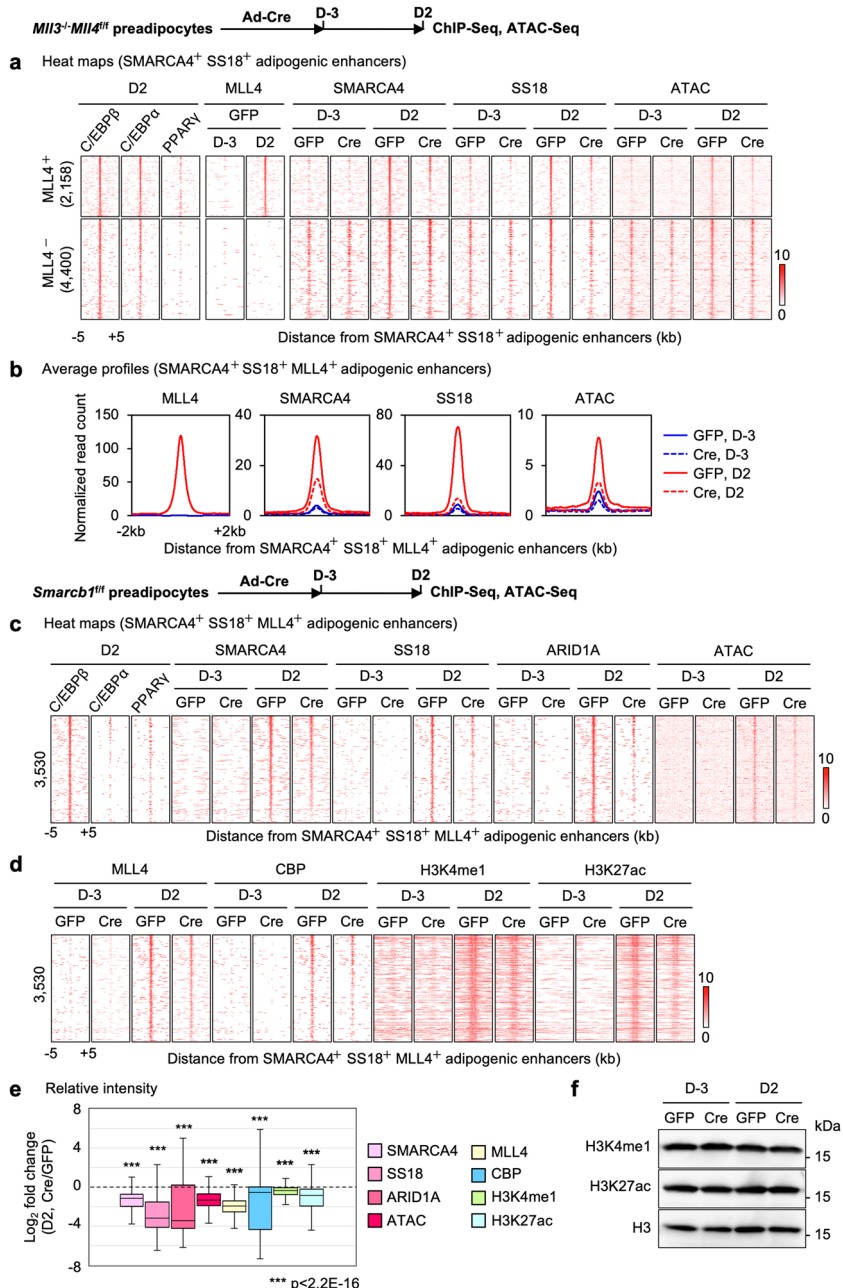

**Fig. 5 Reciprocal regulation between BAF and MLL4 on active enhancers during adipogenesis.** *Mll3*$^{-/-}$*Mll4*$^{f/f}$ or *Smarcb1*$^{f/f}$ brown preadipocytes were infected with adenoviruses expressing GFP or Cre, followed by adipogenesis assays. Cells were collected before (D-3) and during (D2) adipogenesis for ChIP-Seq, ATAC-Seq and Western blot analyses. **a, b** Decreased SMARCA4 and SS18 binding and chromatin accessibility on adipogenic enhancers in *Mll4* KO (Cre) cells during adipogenesis. Adipogenic enhancers were defined as active enhancers bound by either C/EBPβ, C/EBPα, or PPARγ at D2 of adipogenesis. Published ChIP-Seq data sets for C/EBPβ, C/EBPα, and PPARγ were used (GSE74189)[18]. Heat maps around SMARCA4$^{+}$ SS18$^{+}$ adipogenic enhancers (**a**) and average profiles around SMARCA4$^{+}$ SS18$^{+}$ MLL4$^{+}$ adipogenic enhancers (**b**) are shown. Adipogenic enhancers shown in the heat maps were ranked by the binding intensity of SMARCA4 at the center in D2 control (GFP) cells. **c–f** Deletion of *Smarcb1* reduces binding of BAF subunits (SMARCA4, SS18, and ARID1A) as well as chromatin opening, and impairs MLL4 binding and activation of adipogenic enhancers. Adipogenic enhancers at D2 of adipogenesis were re-defined using ChIP-Seq data sets obtained in *Smarcb1*$^{f/f}$ cells. **c** Heat maps for ChIP-Seq of BAF subunits (SMARCA4, SS18, and ARID1A) and ATAC-Seq around the center of SMARCA4$^{+}$ SS18$^{+}$ MLL4$^{+}$ adipogenic enhancers. **d** Heat maps for ChIP-Seq of MLL4, CBP, H3K4me1 and H3K27ac on adipogenic enhancers. Adipogenic enhancers shown in the heat maps were ranked by the intensity of SMARCA4 at the center in D2 control (GFP) cells. **e** Fold changes of intensities between control (GFP) and *Smarcb1* KO (Cre) cells on adipogenic enhancers (*n* = 3530) are shown in box plots. The box represents the first and third quartiles with the horizontal line showing the median, and whiskers indicating minimum and maximum in all box plots. Outliers were not included. Statistical significance levels are as follows (Wilcoxon signed rank test, one-sided): SMARCA4 (*p* = 0); SS18 (*p* = 0); ARID1A (*p* = 6.3E-101); ATAC (*p* = 0); MLL4 (*p* = 4.5E-281); CBP (*p* = 0); H3K4me1 (*p* = 0); H3K27ac (*p* = 0). Exact p-values cannot be computed for SMARCA4, SS18, ATAC, CBP, H3K4me1 and H3K27ac due to ties in the ranks. **f** Western blot analyses of H3K4me1 and H3K27ac in control and *Smarcb1* KO cells during adipogenesis. Total histone H3 was used as a loading control. *n* = 3 biological replicates.

ChIP-Seq revealed that deletion of *Mll4* prevented genomic binding of SMARCA4 and SS18 on MLL4+ adipogenic enhancers but had little effect on MLL4−adipogenic enhancers. Consistently, ATAC-Seq showed that chromatin accessibility at MLL4+ enhancers was also significantly reduced in *Mll4* KO cells (Fig. 5a, b). Reduced binding of BAF on adipogenic enhancers in *Mll4* KO cells was confirmed on *Pparg* and *Fabp4* gene loci (Supplementary Fig. 6b, c).

We next evaluated whether BAF contributes to the genomic binding of MLL4 on adipogenic enhancers during differentiation. ChIP-Seq and ATAC-Seq revealed that *Smarcb1* deletion in preadipocytes reduced genomic binding of SMARCA4, SS18 and ARID1A, as well as chromatin accessibility, on adipogenic enhancers at D2 of adipogenesis (Fig. 5c, e). Moreover, deletion of *Smarcb1* reduced binding of MLL4 and CBP on adipogenic enhancers, indicating that BAF is required for the activation of cell type-specific enhancers (Fig. 5d, e). H3K4me1 and H3K27ac levels also decreased on adipogenic enhancers in *Smarcb1* KO cells, although global levels of these enhancer marks were unchanged (Fig. 5f). Reduced binding of MLL4 on adipogenic enhancers in *Smarcb1* KO cells was confirmed on the *Pparg* gene locus (Supplementary Fig. 6d). Together, these findings reveal reciprocal regulation between BAF and MLL4 on adipogenic enhancers and suggest that BAF and MLL4 coordinate to activate cell type-specific enhancers during differentiation.

**BAF and MLL4 reciprocally promote each other's binding to maintain active enhancers in preadipocytes**. Because we observed interdependent genomic binding of BAF and MLL4 during adipogenesis, we next wondered whether such a relationship is also present in undifferentiated preadipocytes. To examine the genomic association of BAF and MLL4 on AEs in undifferentiated cells, we identified 6,304 BAF+ MLL4+ AEs that were shared by *Mll3*−/−*Mll4*f/f and *Smarcb1*f/f brown preadipocytes and 6,484 BAF+ MLL4−AEs. Deletion of *Mll4* in *Mll3*−/−*Mll4*f/f brown preadipocytes reduced ARID1A binding and H3K27ac levels, but not SMARCA4 binding, on BAF+ MLL4+ AEs in preadipocytes (Fig. 6a, b). Interestingly, the requirement for MLL4 was also observed on MLL4− AEs, suggesting that deletion of *Mll4* has both primary and secondary effects on maintaining BAF genomic binding in preadipocytes. Next, we investigated whether BAF is required for maintaining MLL4 binding on AEs in preadipocytes. While deletion of *Smarcb1* had little effect on maintaining genomic binding of SMARCA4 or chromatin accessibility, it reduced MLL4 binding on BAF+ MLL4+ AEs (Fig. 6c, d). Decreased MLL4 binding on BAF+ MLL4+ AEs was also observed in SMARCA4-depleted preadipocytes in addition to widespread reduction in chromatin accessibility (Fig. 6e, f). Together, these data suggest a reciprocal regulation between BAF and MLL4 in promoting their bindings on AEs in undifferentiated cells.

**MLL4 is required for BAF binding on C/EBPβ-activated enhancers**. Interdependent genomic binding of BAF and MLL4 on AEs during adipogenesis could be due to indirect consequences of differentiation defect and failed induction of LDTFs. To evaluate the primary effect of MLL4 on LDTF-activated enhancers, we used *Mll3*−/−*Mll4*f/f preadipocytes expressing ectopic C/EBPβ, which is a pioneer factor that activates a subset of adipogenic enhancers in preadipocytes without inducing differentiation[14,25]. Neither ectopic expression of C/EBPβ nor deletion of *Mll4* affected expression levels of BAF subunits SMARCA4, SS18 and ARID1A in preadipocytes (Fig. 7a). We identified 571 C/EBPβ-activated enhancers that were bound by MLL4 and BAF subunits SMARCA4 and ARID1A (see Materials and Methods). Among these enhancers, 324 displayed ectopic C/EBPβ-induced de novo

BAF binding (BAF-de novo), whereas the remaining 247 were pre-marked by BAF in control cells before ectopic expression of C/EBPβ (BAF-prebound) (Fig. 7b). As expected, motifs of C/EBPs were enriched on both BAF-de novo and BAF-prebound C/EBPβ-activated enhancers (Fig. 7c). Consistent with our previous finding, deletion of *Mll4* markedly decreased H3K27ac levels on C/EBPβ-activated enhancers. Importantly, deletion of *Mll4* not only prevented C/EBPβ-induced de novo BAF binding (as measured by SMARCA4 and ARID1A), but also compromised genomic binding of SMARCA4 and ARID1A on BAF-prebound enhancers, despite little changes in C/EBPβ binding (Fig. 7d–f). These results indicate that MLL4 is required for genomic binding of BAF on C/EBPβ-activated enhancers and suggest a role of MLL4 in mediating BAF localization on enhancers.

**BAF regulates MLL4 binding on C/EBPβ-activated enhancers**. We next asked whether BAF is required for MLL4 binding on C/EBPβ-activated enhancers. For this purpose, we infected immortalized *Smarca4*AID/AID brown preadipocytes with retroviruses expressing C/EBPβ followed by retroviruses expressing Tir1. Cells were then treated with auxin for 4 h to deplete endogenous SMARCA4 protein. Western blot showed that acute depletion of SMARCA4 did not affect the expression levels of ectopic C/EBPβ in preadipocytes (Fig. 8a). Among 449 C/EBPβ-activated enhancers that we identified, C/EBPβ binding was unchanged at 220 AEs (SMARCA4-independent), while C/EBPβ occupancy was reduced at the remaining 229 AEs (SMARCA4-dependent) upon SMARCA4 depletion (Fig. 8b). Only SMARCA4-independent C/EBPβ binding AEs were enriched with C/EBP motifs while SMARCA4-dependent ones were enriched with motifs of AP1 family TFs, suggesting that SMARCA4-independent C/EBPβ binding on AEs is direct while SMARCA4-dependent C/EBPβ binding is through tethering to AP1-bound AEs (Fig. 8c). To assess the direct effect of SMARCA4 on MLL4 recruitment to C/EBPβ-activated enhancers, we focused on SMARCA4-independent AEs. Despite intact C/EBPβ binding, SMARCA4 depletion led to decreased MLL4 binding and chromatin accessibility on these AEs (Fig. 8d, e). These data demonstrate that SMARCA4 is required for MLL4 binding on C/EBPβ-activated enhancers. Together, our data suggest an interdependent relationship between BAF and MLL4 in pioneer LDTF-mediated enhancer activation (Fig. 8f).

## Discussion

Using adipogenesis and the pioneer factor C/EBPβ-mediated enhancer activation as model systems, we provide several lines of evidence to support that BAF cooperates with MLL4 to promote LDTF-dependent enhancer activation and cell type-specific gene expression during cell differentiation. We found that BAF, similar to MLL4, is required for adipogenesis by using conditional knockout mice and derived cells. BAF, but not PBAF, colocalizes with LDTFs and MLL4 on cell type-specific enhancers. Similar to MLL4, BAF is essential for enhancer activation and cell type-specific gene induction during adipogenesis. By depleting BAF subunits SMARCA4 and SMARCB1 as well as MLL4 in cells, we demonstrated that BAF and MLL4 reciprocally regulate one another's binding on AEs before and during adipogenesis. Finally, by evaluating pioneer factor C/EBPβ-activated enhancers without inducing differentiation, we showed direct evidence for an interdependent relationship between BAF and MLL4 on activating cell type-specific enhancers.

**Function of SWI/SNF complexes in adipogenesis**. Several SWI/SNF subunits have been shown to regulate adipogenesis in culture[28]. Ectopic expression of dominant-negative SMARCA4 interferes with PPARγ-, C/EBPα- and C/EBPβ-stimulated

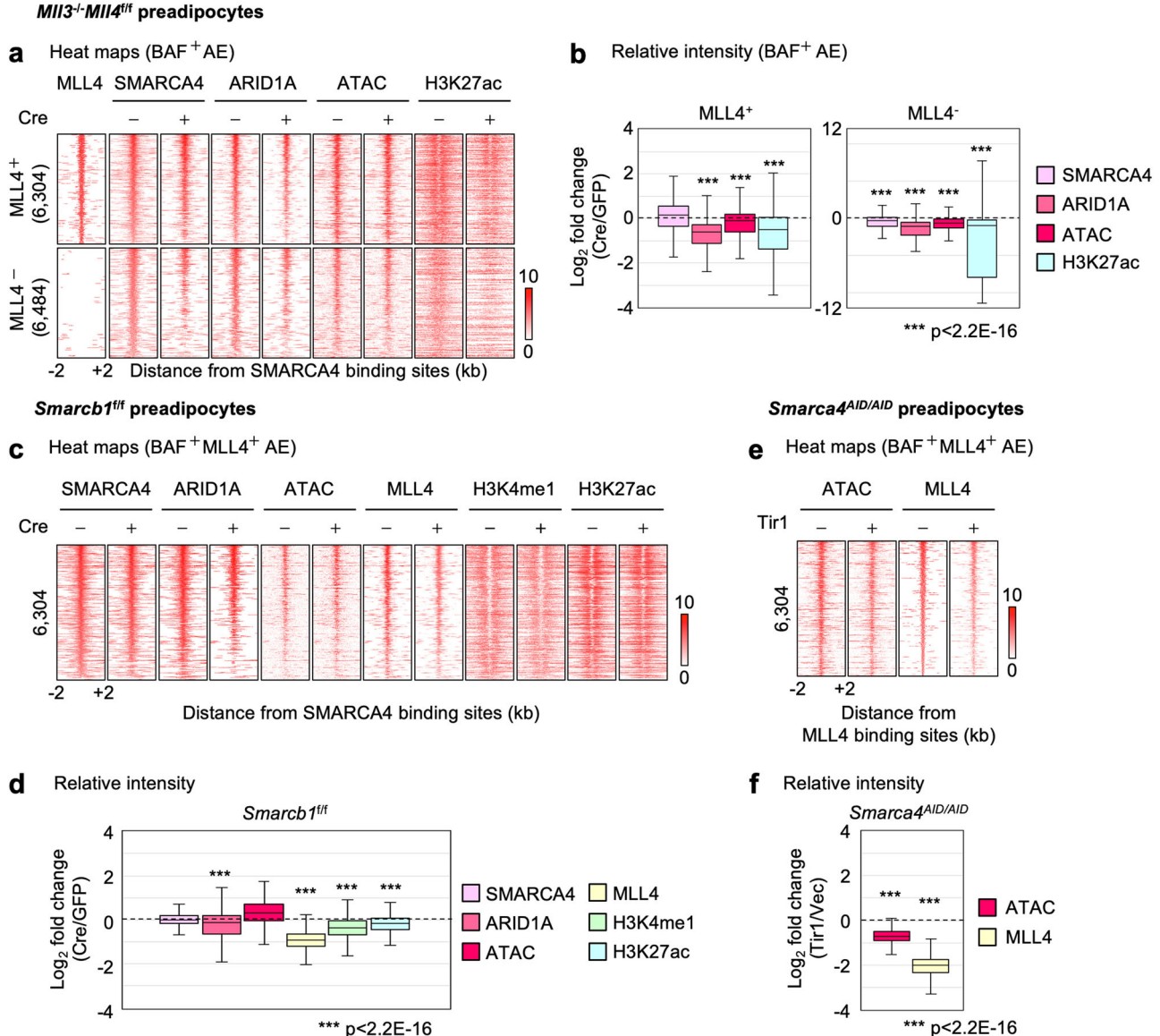

**Fig. 6 BAF and MLL4 reciprocally promote each other's binding to maintain active enhancers in preadipocytes.** *Mll3⁻/⁻Mll4*[f/f] or Smarcb1[f/f] brown preadipocytes were infected with adenoviruses expressing GFP or Cre. Sub-confluent cells were collected for ChIP-Seq and ATAC-Seq. BAF⁺ active enhancers (AEs) were defined as those bound by SMARCA4 and ARID1A. **a, b** Deletion of *Mll4* reduces BAF binding on BAF⁺ AEs in preadipocytes. **a** AEs shown in the heat maps were ranked by the intensity of SMARCA4 at the center in control (GFP) cells. **b** Fold changes of intensities between control and *Mll4* KO (Cre) cells on MLL4⁺ ($n = 6304$) and MLL4⁻ ($n = 6484$) AEs in preadipocytes are shown in box plots. Statistical significance levels are as follows (Wilcoxon signed rank test, one-sided): SMARCA4 ($p = 1$); ARID1A ($p = 0$); ATAC ($p = 5.2E-38$); H3K27ac ($p = 1.6E-293$). Exact p-values cannot be computed for SMARCA4 and ARID1A due to ties in the ranks. **c, d** SMARCB1 is required for maintaining MLL4 binding on BAF⁺ MLL4⁺ AEs in preadipocytes. **c** AEs shown in the heat maps were ranked by the intensity of SMARCA4 at the center in control cells. **d** Fold changes of intensities between control (GFP) and *Smarcb1* KO (Cre) cells on AEs ($n = 6,304$) in preadipocytes are shown in box plots. Statistical significance levels are as follows (Wilcoxon signed rank test, one-sided): SMARCA4 ($p = 1$); ARID1A ($p = 4.0E-59$); ATAC ($p = 1$); MLL4 ($p = 0$); H3K4me1 ($p = 0$); H3K27ac ($p = 1.1E-274$). Exact p-values cannot be computed for SMARCA4, MLL4 and H3K4me1 due to ties in the ranks. **e, f** SMARCA4 is required for maintaining MLL4 binding on BAF⁺ MLL4⁺ AEs in preadipocytes. *Smarca4*[AID/AID] preadipocytes were infected with retroviruses expressing Tir1 or vector (Vec) only, followed by auxin (0.5 mM) treatment for 4 h. Cells were then collected for ATAC-Seq and ChIP-Seq of MLL4. **e** AEs shown in the heat maps were ranked by the intensity of MLL4 at the center in control cells. **f** Fold changes of intensities between SMARCA4-depleted (Tir1) and control (Vec) cells on AEs ($n = 6,304$) in preadipocytes are shown in box plots. Statistical significance levels are as follows (Wilcoxon signed rank test, one-sided): ATAC ($p = 0$); MLL4 ($p = 0$). Exact p-values cannot be computed due to ties in the ranks. In **b**, **d** and **f**, box represents the first and third quartiles with the horizontal line showing the median, and whiskers indicating minimum and maximum in all box plots. Outliers were not included.

adipogenesis in fibroblasts[24]. Knockdown of *Smarcb1* in 3T3-L1 white preadipocytes inhibits adipogenesis[31]. Using the AID system in brown preadipocytes, we demonstrated that endogenous SMARCA4 is essential for adipogenesis. We showed the essential role of SMARCB1 in adipogenesis of SV40T-immortalized brown

preadipocytes. Immortalization by SV40T, which inactivates tumor suppressors p53 and Rb[32], prevents growth defects caused by *Smarcb1* deletion[33,34], and enables the study of SMARCB1 in preadipocyte differentiation. Further, we showed that SMARCB1 is required for brown adipose tissue development in mice We also

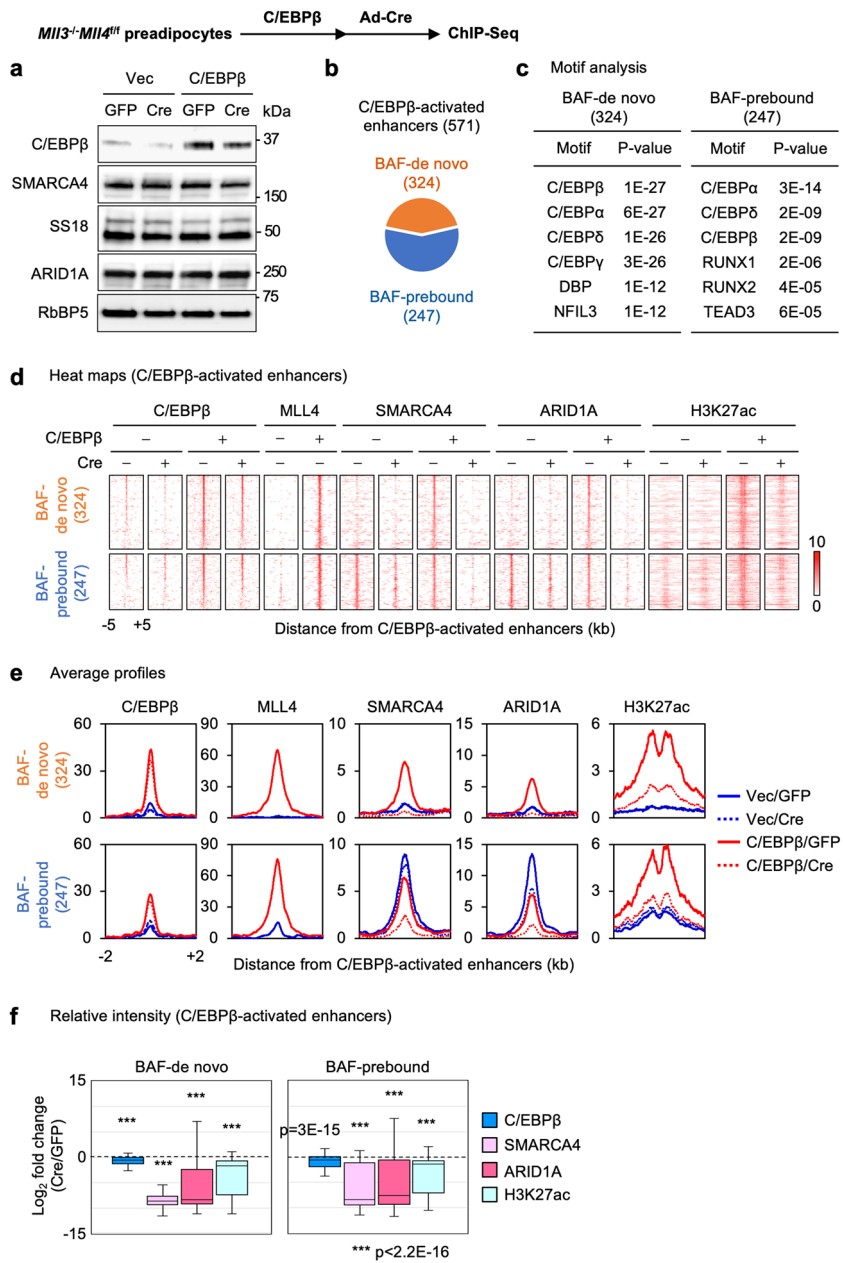

**Fig. 7 MLL4 is required for BAF binding on C/EBPβ-activated enhancers.** *Mll3⁻/⁻Mll4^f/f* brown preadipocytes were infected with retroviruses expressing C/EBPβ or vector (Vec), and then infected with adenoviruses expressing GFP or Cre. Sub-confluent cells were collected without inducing adipogenesis for Western blot and ChIP-Seq analyses. Published ChIP-Seq data sets for C/EBPβ, MLL4, H3K27ac were used (GSE50466)[14]. **a** Western blot analyses of BAF subunits (SMARCA4, SS18, and ARID1A) and C/EBPβ expression in control (GFP) and *Mll4* KO (Cre) preadipocytes. n = 2 biological replicates. **b** Among the 571 C/EBPβ-activated enhancers in preadipocytes, BAF (SMARCA4 and ARID1A) binding was induced on 324 AEs only after ectopic expression of C/EBPβ (BAF-de novo), while 247 AEs were already bound by BAF before ectopic expression of C/EBPβ in preadipocytes (BAF-prebound). **c** Motif analysis of BAF-de novo and BAF-prebound AEs using SeqPos motif tool. **d**–**f** Deletion of *Mll4* reduces BAF binding on C/EBPβ-activated enhancers. Heat maps (**d**) and average profiles (**e**) around C/EBPβ-activated enhancers are shown. Enhancers shown in the heat maps were ranked by the intensity of C/EBPβ at the center in control cells expressing ectopic C/EBPβ. **f** Fold changes of intensities between control and *Mll4* KO (Cre) cells on BAF-de novo (n = 324) and BAF-prebound (n = 247) C/EBPβ-activated enhancers are shown in box plots. The box represents the first and third quartiles with the horizontal line showing the median, and whiskers indicating minimum and maximum in all box plots. Outliers were not included. Statistical significance levels are as follows (Wilcoxon signed rank test, one-sided): BAF-de novo C/EBPβ (p = 1.3E-19); BAF-de novo SMARCA4 (p = 1.1E-51); BAF-de novo ARID1A (p = 4.1E-50); BAF-de novo H3K27ac (p = 1.1E-48); BAF-prebound C/EBPβ (p = 2.9E-15); BAF-prebound SMARCA4 (p = 1.0E-32); BAF-prebound ARID1A (p = 3.4E-27); BAF-prebound H3K27ac (p = 5.6E-34).

observed that the BAF-specific subunit ARID1A is required for adipogenesis in preadipocytes. Together, these findings demonstrate a critical role of BAF in adipogenesis. Our observations that PBAF-specific PBRM1 is dispensable for adipogenesis in vitro and in vivo provide an example of distinct functions of BAF and

PBAF in regulating cell differentiation and development. The role of GBAF in adipogenesis remains to be determined.

It has been shown that over 90% of mature brown adipocytes arise from Myf5-Cre expressing precursors in mice[35,36]. However, since the *Smarcb1* gene is located on the same chromosome 10 as

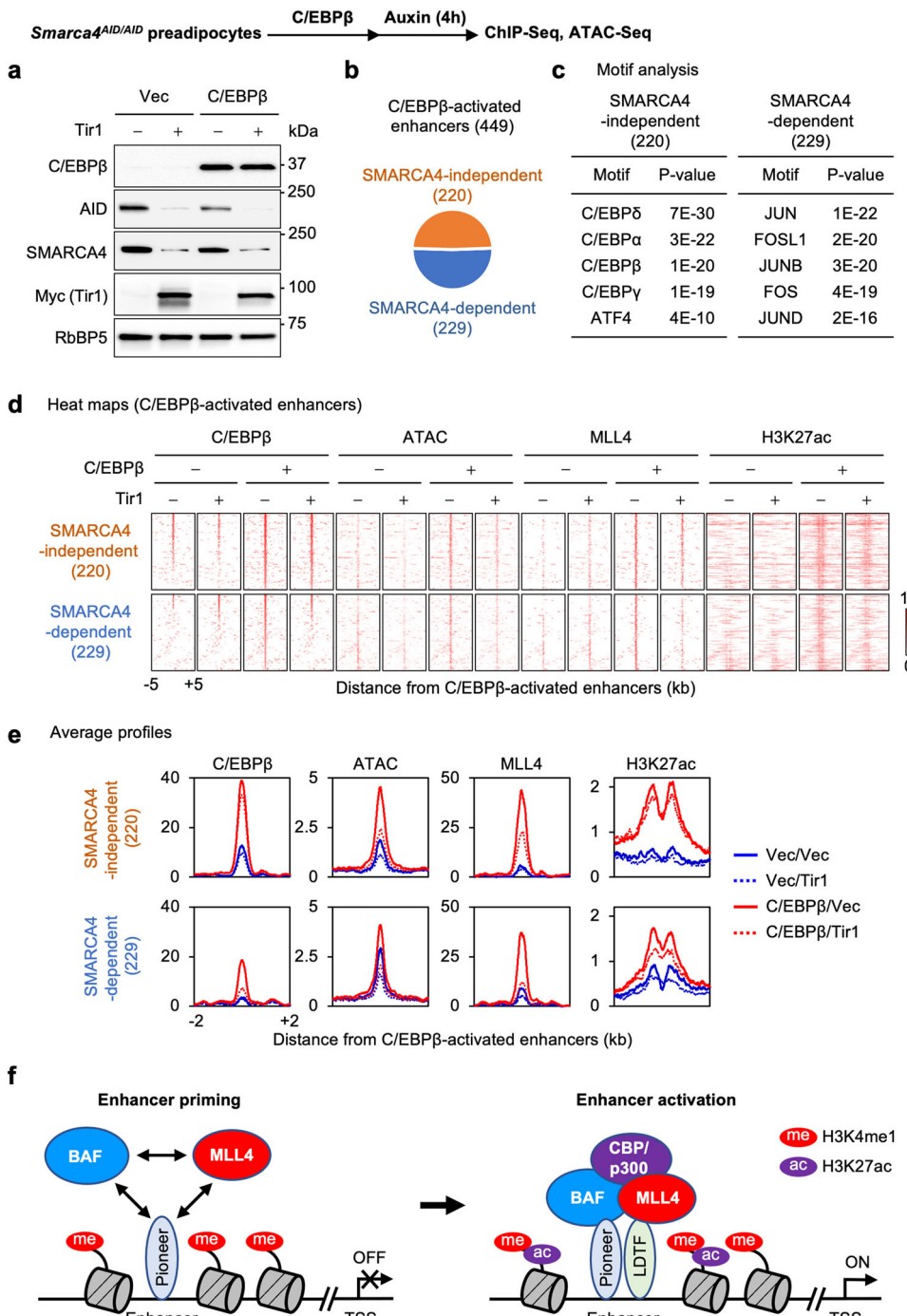

**Fig. 8 BAF regulates MLL4 binding on C/EBPβ-activated enhancers.** *Smarca4*[AID/AID] brown preadipocytes were infected with retroviruses expressing C/EBPβ or vector (Vec), and then infected with retroviruses expressing Tir1. Sub-confluent cells were treated with auxin for 4 h and collected for Western blot, ChIP-Seq and ATAC-Seq analyses. **a** Western blot analyses using antibodies indicated on the left. $n = 3$ biological replicates. **b** Among the 449 C/EBPβ-activated enhancers in preadipocytes, 220 exhibited SMARCA4-independent C/EBPβ binding, while the remaining 229 showed SMARCA4-dependent C/EBPβ binding. **c** Motif analysis of SMARCA4-independent and -dependent C/EBPβ-activated enhancers in preadipocytes using SeqPos motif tool. **d, e** Auxin-induced, Tir1-dependent acute depletion of SMARCA4 reduces MLL4 binding on C/EBPβ-activated enhancers in preadipocytes. Heat maps (**d**) and average profiles (**e**) around C/EBPβ-activated enhancers are shown. Enhancers shown in the heat maps were ranked by the intensity of C/EBPβ at the center in control cells expressing ectopic C/EBPβ. **f** A model depicting the interdependent relationship between BAF and MLL4 in promoting pioneer TF- and LDTF-dependent cell type-specific enhancer activation.

the *Myf5-Cre* transgene, there is a technical problem of low recombination frequency (1/24, according to the distance between two loci). We instead used *PdgfRa-Cre* to delete *Smarcb1* in precursor cells of adipocytes as described previously[27]. *PdgfRa-Cre* is also expressed in other cell types including cranial and

cardiac neural crest cells[37]. Neonatal death of *Smarcb1*[f/f];*PdgfRa-Cre* is likely caused by *Smarcb1* deletion in cranial neural crest cells, which led to defects in coronal sutures, but not by *Smarcb1* deletion in brown adipose precursors as complete absence of BAT does not affect survival of laboratory mice[38].

**Distinct genomic localization of BAF, PBAF and GBAF**. Using adipogenesis as a model system for cell differentiation, we confirmed previous observations on the distinct genomic localization of BAF, PBAF and GBAF in ESCs and in cancer cells: BAF to AEs, PBAF to promoters and GBAF to CTCF binding sites[11,12,39]. Although SMARCA4 and SS18 were reported to physically associate with both BAF and GBAF complexes, we found that on AEs, the vast majority of them are associated with BAF. We also showed that BAF is the major SWI/SNF complex on AEs and that BAF binding sites dynamically change during adipogenesis. Motifs of AP-1 family TFs are enriched on BAF-bound AEs in preadipocytes, consistent with a recent finding that AP-1 can recruit BAF to establish accessible chromatin in fibroblasts[40]. Although GBAF is enriched on CTCF binding sites, it also colocalizes with BAF on a subset of AEs during adipogenesis. The role of GBAF on AEs in cell differentiation remains unclear.

**Reciprocal regulation between BAF and MLL4 in enhancer activation**. We identified 4,410 BAF-prebound AEs and 6,403 BAF-de novo AEs at D2 of adipogenesis (Fig. 3d, e). While BAF-prebound enhancers were constitutively active, BAF-de novo enhancers were activated from D-3 (preadipocytes) to D2 of adipogenesis. BAF-de novo AEs were more enriched with MLL4 binding (63.4%; 4,065/6,403), compared with BAF-prebound ones (6.7%; 297/4,410) at D2 of adipogenesis. In contrast to MLL4⁻ AEs, MLL4⁺ ones showed markedly increased binding of BAF and LDTFs, chromatin accessibility and enhancer activation during adipogenesis (Supplementary Fig. 5). These observations suggest that BAF cooperates with MLL4 to activate cell type-specific enhancers during cell fate transition.

It was reported that H3K4me1-marked nucleosome can pull down SWI/SNF components including SMARCA4 from HeLa nuclear extract and that disruption of catalytic activities of MLL3/4 reduces SMARCA4 binding on H3K4me1⁺ distal regions in undifferentiated mouse embryonic stem cells[41]. On the other hand, it was also shown that BAF (SMARCA4 and SS18) does not require pre-marked H3K4me1 for binding and that targeting of ectopic SMARCA4 in SMARCA4-deficient cancer cells establishes MLL3/4 genomic binding and active enhancer marks de novo[42]. Another study showed that UTX, a component of MLL3/MLL4 complex, physically associates with SMARCA4 in mouse myeloid cells and that loss of UTX leads to changes in chromatin accessibility and enhancer activation[43]. Through the use of several gene knockout and protein depletion systems, we establish a bidirectional relationship between BAF and MLL4 on AEs in preadipocytes and during adipogenesis, which rectifies seemingly conflicting results from previous separate studies. Reciprocal regulation between BAF and MLL4 is also supported by their physical association observed in this study and reported in the literature[44]. Whether BAF and MLL4 regulate each other on AEs through physical interaction of their components or enhancer histone marks remains to be determined.

**Interplay of BAF and MLL4 promotes pioneer factor-dependent enhancer activation**. Pioneer factors such as OCT4 and C/EBPβ physically associate with SMARCA4 and other BAF subunits in cells[23,45]. Pioneer factors recruit SMARCA4 to induce chromatin accessibility to facilitate enhancer activation[42]. We reported previously that OCT4 and C/EBPβ physically associate with the MLL4 complex in cells and that ectopic C/EBPβ recruits MLL4 to activate a subset of C/EBPβ binding adipogenic enhancers in undifferentiated cells[14,20]. It is worth noting that MLL4 loss has little effect on C/EBPβ binding on AEs. In contrast, acute depletion of SMARCA4 reduces C/EBPβ occupancy on AEs, consistent with recent reports on the role of BAF in the

binding of pioneer factors on enhancers[46–49]. Our findings in the current study further suggest a model for reciprocal regulation between BAF and MLL4 on C/EBPβ-activated enhancers. The pioneer factor C/EBPβ recruits BAF to remodel chromatin to enable MLL4 binding on enhancers. Meanwhile, C/EBPβ recruits MLL4 to facilitate BAF binding on enhancers. BAF and MLL4 may cooperate with C/EBPβ to maintain chromatin accessibility and active enhancer landscape and facilitate other LDTFs binding (Fig. 8f). Thus, BAF and MLL4 form a positive feedback loop to activate enhancers and lineage-specific gene expression during cell differentiation.

## Methods

**Plasmids and antibodies**. The retroviral plasmids pBABEneo-SV40LT and pWZLhygro-C/EBPβ have been described[14]. The homemade antibodies anti-UTX[50] and anti-MLL4[51] have been described. Anti-SMARCB1/SNF5/INI1 (A-5, sc-166165), anti-SMARCC2/BAF170 (E-6, sc-17838X), anti-ARID2/BAF200 (E-3, sc-166117X), anti-C/EBPβ (C-19, sc-150X), anti-C/EBPα (144AA, sc-61X) and anti-PPARγ (H-100, sc-7196X) were from Santa Cruz. Anti-RbBP5 (A300-109A) was from Bethyl Laboratories. Anti-SMARCA4/BRG1 (ab110641) and anti-H3K27ac (ab4729) were from Abcam. Anti-ARID1A/BAF250A (D2A8U, #12354), anti-SS18 (D6I4Z, #21792) and anti-CBP (D6C5, #7389) were from Cell Signaling. Anti-BRD9 (#61537) was from Active Motif. Anti-H3K4me1 (13-0040) was from EpiCypher.

**Mouse experiments**. To generate Smarcb1[f/f];PdgfRα-Cre mice, Smarcb1[f/f] mice[52] obtained from Charles W.M. Roberts (St. Jude Children's Research Hospital, Memphis, TN) were crossed with PdgfRα-Cre mice (Jackson No.013148). Pbrm1[f/f] mice (Jackson No. 029049) were crossed with Myf5-Cre mice (Jackson No. 007893) to get Pbrm1[f/f];Myf5-Cre mice. Generation of Smarca4[AID/AID] mice will be described in a separate manuscript. Mice were housed in a room with controlled temperature of 22 °C and humidity of 45–65% under a 12-h light and 12-h dark cycle. Histology and immunohistochemistry analyses of E18.5 embryos were done as described[14]. Anti-Myosin (1:20 dilution, MF20; Developmental Studies Hybridoma Bank) and anti-UCP1 (1:400 dilution, ab10983; Abcam) were used. For fluorescent secondary antibodies, anti-mouse Alexa Fluor 488 and anti-rabbit Alexa Fluor 555 (Life Technologies, Carlsbad, CA, USA) were used. All mouse work was approved by the Animal Care and Use Committee of NIDDK, NIH.

**Isolation and immortalization of primary brown preadipocytes, virus infection and adipogenesis assay**. Primary brown preadipocytes were isolated from the interscapular BAT of newborn pups and were immortalized with retroviral vectors expressing SV40T as described[51]. Adenoviral infection of preadipocytes was done at 50 moi. Cells were routinely cultured in DMEM plus 10% FBS. For adipogenesis assays, preadipocytes were plated at a density of $1 \times 10^5$ in each well of 6-well plates in growth medium (DMEM plus 10% FBS) 4 days before induction of adipogenesis. At day 0, cells were fully confluent and were treated with differentiation medium (DMEM plus 10% FBS, 0.1 μM insulin and 1 nM T3) supplemented with 0.5 mM IBMX, 1 μM DEX, and 0.125 mM indomethacin. Two days later, cells were changed to the differentiation medium. Cells were replenished with fresh medium at 2-day intervals. Fully differentiated cells were either stained with Oil Red O or subjected to gene expression analyses.

**The auxin-inducible degron (AID) system for SMARCA4 depletion**. Smarca4[AID/AID] preadipocytes were isolated from interscapular BAT of Smarca4[AID/AID] newborn pups and immortalized with SV40T. Immortalized Smarca4[AID/AID] preadipocytes were infected with retroviral vector expressing the auxin receptor Tir1 of rice (pBabePuro-OsTir1-9Myc, Addgene #80074). To induce degradation of SMARCA4, auxin (indole-3-acetic acid sodium salt, Sigma-Aldrich #I5148) was added to the culture medium in a final concentration of 0.5 mM for indicated times.

**shRNA knockdown**. For shRNA transduction, individual shRNAs targeting mouse Arid1a (TRCN0000238303, shArid1a-#1; TRCN0000238306, shArid1a-#5) were obtained from Sigma Mission shRNA library (Sigma-Aldrich). Immortalized Smarcb1[f/f] brown preadipocytes were infected with lentiviral shRNA targeting Arid1a or control virus alone for 24 h. Infected cells were selected with puromycin (2 μg/mL) for 4 days before performing further experiments.

**Western blot and immunoprecipitation**. Nuclear proteins were extracted using the nuclear extract preparation method as described previously[19]. Briefly, cells were washed with cold PBS, resuspended in buffer A (10 mM HEPES, pH 7.9, 1.5 mM MgCl₂, 10 mM KCl and 0.1% NP40) supplemented with protease inhibitors (Roche), 0.5 mM DTT and 0.2 mM phenylmethylsulfonyl fluoride (PMSF), and incubated on ice for 10 min. After centrifugation at 1,000 g, nuclei were resuspended in buffer C (20 mM HEPES, pH 7.9, 1.5 mM MgCl₂, 420 mM NaCl, 0.2 mM EDTA and 25% glycerol) supplemented with protease inhibitors, 0.5 mM DTT

and 0.2 mM PMSF. Nuclear extracts were separated using 4–15% Tris-Glycine gradient gels (Bio-Rad Laboratories). For Western blotting of MLL4, nuclear extracts were separated using 3–8% Tris-Acetate gradient gels (Invitrogen). Total proteins on the gel were transferred to a PVDF membrane (Bio-Rad Laboratories). The membranes were probed using specific antibodies.

For co-immunoprecipitation (IP), nuclear proteins were extracted from HEK293T and preadipocytes, diluted with an IP Buffer (50 mM Tris-HCl, pH 7.5, 1 mM EDTA, 150 mM NaCl, and 1% Triton X-100) supplemented with protease inhibitors, 1 mM DTT and 1 mM PMSF, and incubated with anti-MLL4, anti-UTX, anti-SMARCA4 or anti-SMARCB1 antibodies overnight at 4 °C. Samples were additionally incubated with Dynabeads Protein A (Life Technologies, 10008D) at 4 °C for 2 h. Beads were washed three times with an IP buffer and the immunoprecipitates were eluted with 20 μL of 1x sample buffer (NuPAGE LDS buffer, Thermo Scientific) including 100 mM DTT.

**IP-Mass spectrometry**. The immunoprecipitation was carried out as described[53]. Briefly, the nuclear extract was prepared from undifferentiated mouse ES cells and immunoprecipitated with an antibody against MLL4 or UTX. The eluted immunoprecipitates were subjected to mass spectrometric analyses by Taplin Biological Mass Spectrometry Facility (Harvard Medical School). Peptides were separated on a HPLC column (100μm internal diameter, 25 cm in length) packed with 5μm Magic C18AQ 200 A (Michrom Bioresources). A LC gradient of Buffer A (97% water, 3% acetonitrile, 0.1% formic acid) and Buffer B (97% acetonitrile, 3% water, 0.1% formic acid) were used for separation, from 5% to 30% of B over one hour. For gas phase fragmentation settings, CID fragmentation with 2 m/z isolation width and 35% normalized collision energy were used. MS/MS scans were performed on a stand-alone LTQ velos which only acquired data at low resolution. SEQUEST (ver. 28, rev. 13) was used for peptide identification. Protein sequence databases were downloaded from Uniprot. The database was indexed allowing for partially tryptic peptides at either terminal end. Two missed internal cleavages were allowed. A fixed modification was on Cysteine (71.037 Da) for mono-acrylamide adducts. The only variable modification was on Methionine (15.99) for oxidation. The precursor mass tolerance was 2 Da, and 1 Da for fragment ions. The peptide false discovery rate (FDR) for sample UTX IP sample was 0.29% and the protein FDR was 3.35%. The peptide FDR for MLL4 IP sample was 0.20% and the protein FDR was 2.09%. Two or more unique peptides for protein identification were considered to have confidence.

**RNA isolation and qRT-PCR analysis**. Total RNA was extracted using TRIzol (Life Technologies) and reverse transcribed using ProtoScript II first-strand cDNA synthesis kit (NEB, E6560), following manufacturer's instructions. Quantitative RT-PCR (qRT-PCR) was performed with the Luna® Universal qPCR Master Mix (NEB, M3003) using QuantStudio 5 Real-Time PCR System (Thermo Fisher). PCR amplification parameters were 95 °C (3 min) and 40 cycles of 95 °C (15 s), 60 °C (60 s), and 72 °C (30 s). Primer sequences are provided in Supplementary Table 1. Statistical significance was calculated using the two-tailed unpaired $t$-test on two experimental conditions.

**RNA-Seq library preparation**. Total RNA was subjected to mRNA purification using the NEBNext Poly(A) mRNA Magnetic Isolation Module (NEB, E7490). Isolated mRNAs were reverse-transcribed into double stranded cDNA and subjected to sequencing library construction using the NEBNext Ultra™ II RNA Library Prep Kit for Illumina (NEB, E7770) according to the manufacturer's instructions. RNA libraries were sequenced on Illumina HiSeq 3000.

**ChIP and ChIP-Seq library preparation**. Chromatin immunoprecipitation (ChIP) and ChIP-Seq were done as described[54]. Briefly, cells were cross-linked with 1.5% formaldehyde for 10 min and quenched by 125 mM glycine for 10 min. 10 million fixed cells were resuspended in ice cold buffer containing 5 mM PIPES, pH 7.5, 85 mM KCl, 1% NP-40 and protease inhibitors, incubated on ice for 15 min, and centrifuged at 500 g for 5 min at 4 °C. Nuclei were resuspended with 1 mL buffer containing 50 mM Tris-HCl, pH8.0, 10 mM EDTA, 0.1% SDS and protease inhibitors, and subjected to sonication. Sheared chromatin was clarified by centrifugation at 13,000 g for 10 min at 4 °C. The supernatant was transferred to a new tube and further supplemented with 1% Triton-X100, 0.1% sodium deoxycholate and protease inhibitors. 2% of the mixture was set aside as input and 20 ng of spike-in chromatin (Active Motif, #53083) was added to the rest. For each ChIP, 4 − 10 μg of target antibodies and 2 μg of spike-in antibody (anti-H2Av, Active Motif, #61686) were added and incubated on a rotator at 4 °C overnight. ChIP samples were added with 50 μL prewashed protein A Dynabeads (ThermoFisher) and incubated for 3 h at 4 °C. Beads were then collected on a magnetic rack and washed twice with 1 mL cold RIPA buffer, twice with 1 mL cold RIPA buffer containing 300 mM NaCl, twice with 1 mL cold LiCl buffer and once with PBS. Beads were then eluted with 100 μL buffer containing 0.1 M NaHCO₃, 1% SDS, and 20 μg Proteinase K at 65 °C overnight. Input sample volumes were adjusted to 100 μL with the elution buffer. Samples were then purified using QIAquick PCR purification kit (Qiagen) and eluted in 30 μL 10 mM Tris-HCl. For ChIP-Seq, entire ChIPed DNA or 300 ng of the input DNA were used to construct libraries using NEBNext Ultra II DNA Library Prep kit with AMPure XP magnetic beads

(Beckman Coulter). Library quality and quantity were estimated with Bioanalyzer and Qubit assays. The final libraries were sequenced on Illumina HiSeq 3000.

**ATAC-Seq library preparation**. The Assay for Transposase-Accessible Chromatin with high-throughput sequencing (ATAC-Seq) was performed as described[55]. For each ATAC reaction, briefly, $5 \times 10^4$ freshly collected cells were aliquoted into a new tube and spun down at $500 \times g$ for 5 min at 4 °C. The cell pellet was incubated in 50 μL of ATAC-RSB buffer (10 mM Tris-HCl pH 7.4, 10 mM NaCl, 3 mM MgCl₂) containing 0.1% NP-40, 0.1% Tween-20, and 0.01% digitonin (Promega) on ice for 3 min and was washed out with 1 mL of ATAC-RSB buffer containing 0.1% Tween-20. Nuclei were collected by centrifugation at $500 \times g$ for 10 min at 4 °C, then resuspended in 50 μL of transposition reaction buffer containing 25 μL 2× Tagment DNA buffer, 2.5 μL transposase (100 nM final; Illumina), 16.5 μL PBS, 0.5 μL 1% digitonin, 0.5 μL 10% Tween-20, and 5 μL H₂O. The reaction was incubated for 30 min at 37 °C with mixing (1000 r.p.m.) and directly subjected to DNA purification using the MinElute Reaction Cleanup Kit (Qiagen) according to the manufacturer's instructions. Eluted DNA was amplified with PCR using Nextera i7- and i5-index primers (Illumina). Purification and size selection of the amplified DNA were carried out with AMPure XP magnetic beads (Beckman Coulter) to remove primer dimers and >1,000 bp fragments. For purification, the ratio of sample to beads was set to 1:1.8, whereas for size selection, the ratio was set to 1:0.55. Sequencing libraries were analyzed with Qubit and sequenced on HiSeq3000.

### Computational analysis

*RNA-Seq data analysis*. Raw sequencing data were aligned to the mouse genome mm9 using STAR software[56]. Reads on exons were collected to calculate reads per kilobase per million (RPKM) as a measure of gene expression level. Only genes with exonic reads of RPKM > 1 were considered expressed. For comparing gene expression levels before (D-3) and during (D2) adipogenesis, or in control and SMARCB1-deficient cells (Supplementary Fig. 2), fold change cutoff of > 2.5 was used to identify differentially expressed genes. Gene ontology (GO) analysis was done using DAVID with the whole mouse genome as background [https://david.ncifcrf.gov].

*ChIP-Seq peak calling*. Raw sequencing data were aligned to the mouse genome mm9 and the drosophila genome dm6 using Bowtie2 (v2.3.2)[57]. To identify ChIP-enriched regions, SICER (v1.1) was used[58]. For ChIP-Seq of histone modifications (H3K4me1 and H3K27ac), the window size of 200 bp, the gap size of 200 bp, and the false discovery rate (FDR) threshold of $10^{-3}$ were used. For ChIP-Seq of non-histone factors, the window size of 50 bp, the gap size of 50 bp, and the FDR threshold of $10^{-10}$ were used.

*Genomic distribution of ChIP-Seq peaks*. To define regulatory regions (Fig. 1c), combination of genomic coordinates and histone modification ChIP-Seq data were used. Promoter was defined as transcription start sites ±1 kb. Promoter-distal regions were further separated into active enhancers (H3K4me1⁺ H3K27ac⁺), primed enhancers (H3K4me1⁺ H3K27ac⁻), and the other (H3K4me1⁻).

*Motif analysis*. To find enriched TF motifs in given genomic regions identified by ChIP-Seq, we utilized the SeqPos motif tool in the Cistrome toolbox [http://cistrome.org/ap/][59]. For ChIP-Seq data with more than 5,000 regions (Figs. 1, 3 and Supplementary Fig. 4), top 3,000 significant regions were used. Top 3,000 significant peaks were chosen based on the FDR value provided by SICER[58]. Briefly, FDR was calculated using p-value adjusted for multiple testing, and p-value was based on a test using the Poisson distribution. For ChIP-Seq data with small numbers (Figs. 7 and 8), all regions were used. For differential motif analysis of given two regions (Supplementary Fig. 5), HOMER software was used [http://homer.ucsd.edu/homer/][60].

*Heat maps and box plots*. The heat map matrices were generated using in-house scripts with 50 bp resolution and visualized in R using gplots package. Enhancers shown in the heat maps were ranked according to the intensity of SMARCA4 at the center of 400 bp window in control cells (Figs. 5 and 6) or that of C/EBPβ in control cells expressing ectopic C/EBPβ (Figs. 7 and 8). The ratio of normalized ChIP-Seq read counts in SMARCB1-deficient and control cells (Figs. 5e and 6d), those in MLL4⁻-deficient and control cells (Fig. 6b), or those in SMARCA4-depleted and control cells (Fig. 6f) in base 2 logarithm were plotted using box plot, with outliers not shown. Wilcoxon signed rank test (one-sided) was used to determine statistical differences in SMARCB1-deficient and control cells (Figs. 5e and 6d), in MLL4-deficient and control cells (Fig. 6b), or in SMARCA4-depleted and control cells (Fig. 6f).

*ATAC-Seq data processing*. Raw ATAC-Seq reads were processed using Kundaje lab's ataqc pipelines [https://github.com/kundajelab/atac_dnase_pipelines] which included adapter trimming, aligning to mouse mm9 genome by Bowtie2, and peak calling by MACS2. For downstream analysis, we used filtered reads that were retained after removing unmapped reads, duplicates and mitochondrial reads.

*C/EBPβ-activated enhancers*. To identify C/EBPβ-activated enhancers in pre-adipocytes (Figs. 7 and 8), we first selected C/EBPβ+ AEs (C/EBPβ+ H3K27ac+ promoter-distal regions) in ectopic C/EBPβ-expressed cells. We further narrowed the focus down to SMARCA4+ ARID1A+ MLL4+ ones to better assess the direct relationship between BAF and MLL4. We compared H3K27ac enrichment intensities in control and ectopic C/EBPβ-expressed cells. Enhancers with over 2-fold increased H3K27ac intensities in ectopic C/EBPβ-expressed cells were considered as C/EBPβ-activated enhancers.

**Reporting summary**. Further information on research design is available in the Nature Research Reporting Summary linked to this article.

## Data availability

The data that support this study are available from the corresponding author upon reasonable request. All ChIP-Seq, RNA-Seq and ATAC-Seq datasets described in this paper have been deposited in NCBI Gene Expression Omnibus under access #GSE151115 [https://www.ncbi.nlm.nih.gov/geo/query/acc.cgi?acc=GSE151115]. The raw mass spectrometry proteomics data are provided in Supplementary Data 1 and 2. Protein sequence databases were downloaded from Uniprot [https://www.uniprot.org/]. Source data are provided with this paper.

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

## Acknowledgements

We thank Keji Zhao for *Smarca4*[AID/AID] mice, Charles W. M. Roberts for *Smarcb1*[f/f] mice, Ruth Kopyto for assistance in schematic drawing, Guojia Xie for suggestions, NIDDK Genomics Core and NHLBI DNA Sequencing and Genomics Core for next generation sequencing, Ross Tomaino of Taplin Biological Mass Spectrometry Facility (Harvard Medical School) for mass spectrometry. This work was supported by the Intramural Research Program of NIDDK, NIH to K.G.

## Author contributions

Conceptualization, Y.-K.P., J.-E.L. and K.G.; Methodology, Y.-K.P., J.-E.L., Z.Y., W.P. and W.W.; Investigation, Y.-K.P., T. O., K.M. and K.G; Software, Formal Analysis, and Data Curation, J.-E.L. and W.P.; Writing – Original Draft, Y.-K.P., J.-E.L. and K.M.; Writing – Review & Editing, Y.-K.P., J.-E.L., K.M. and K.G.; Project Administration and Funding Acquisition, K.G.

## Funding

## Competing interests

The authors declare no competing interests.
