## [Peer Review File · Nature Communications]

REVIEWER COMMENTS

Reviewer #1 (Remarks to the Author):

Park and colleagues test the functional interplay between the SWI/SNF complex and MLL4 during adipogenesis using several genetic model systems. They present evidence for the requirement of the canonical BAF complex (among the three SWI/SNF complex variants) in adipogenesis. This is demonstrated by a combination of transcriptomic profiling, red oil o staining, and in vivo characterization after acute deletion of Smarca4 (using degron-based system), conditional deletion of Smarcb1, and shRNA-mediated deletion of Arid1a. This requirement is not true of PBAF because conditional deletion of Pbrm1 resulted in no or little defect during adipocyte differentiation. Using ChIP-seq studies, they observed binding of the BAF complex at active enhancers, some of which were pre-bound in pre-adipocytes and others were gained de novo during differentiation. Interestingly, when BAF complex binding sites are categorized based on MLL4 co-localization, the motifs enriched differ such that MLL+ sites are more enriched for adipogenic LDTFs and MLL- sites are more enriched for AP-1 family members. They then probed for the requirement of MLL4 or BAF subunit for reciprocal genomic binding. They report interdependence between binding of MLL4 and the BAF complex in adipocytes and in pre-adipocytes. Finally, they exogenously expressed CEBPb in pre-adipocytes to overcome effects of failed LDTF induction and assessed MLL4 and BAF complex binding, where they report reciprocal interdependence at CEBPb-gained enhancers. This is a nice study and thorough study looking at the interdependence of these factors with reciprocal genetic deletions.

Major Comments:

1. Given that aspects of MLL3/4 activity are independent of their catalytic activity (Dorigi et al Mol Cell 2017), and that the maintenance of SMARCA4 binding was shown to be partially dependent on H3K4me in ESCs (Local Nat Genetics 2018), the authors should test whether SMARCA4/SS18 binding is reduced in MLL4 dCD reconstitution relative to MLL4 WT reconstitution in Mll3/4 dKO cells at gained active enhancers. This would be a nice addition to the model and test whether the mark itself is required for recruitment of BAF complexes in particular to de novo sites.
2. Please include the sequencing depth and number of peaks called for all the ChIP-seq experiments in the Reporting Summary. In addition, some of the heatmaps appear very washed out – please include a scale for all heatmaps.
3. Replicate ChIP-seq datasets should be performed for ARID2 and BRD9 in Figure 1C. Overall, the number of peaks is quite a bit lower in D2 than D-3. It's unclear if this is due to biological or technical variation.
4. It's difficult to assess quality of the mass spectrometry data by looking at hand-picked proteins. How many IP-MS replicates were done and was there an anti-IgG control used? The authors should use programs (SAINT, MaxQuant, etc.) to determine which proteins are statistically enriched in their IPs

versus the IgG controls. Mass spectrometry data also needs to be deposited to a public repository like PRIDE (<https://www.ebi.ac.uk/pride/archive/>).

5. In Figure 4 and 5, it's unclear whether the requirement for MLL4 is specific for MLL4+ sites. MLL4- sites should be included in the heatmap analysis to establish the specific role of MLL4 in stabilizing BAF complex at sites where they are co-localized.

6. In Figure 6D, CEBPb binding appears to be reduced in the CEBPb OE/Mll4-deleted samples compared to CEBPb OE/no cre treatment, despite the fact that the aggregate signal overlays on top of each other (Figure 6E). Please include box plots of log₂fc as done in other figures. Given several previous studies showing the interdependence of TF and BAF complex binding (King and Klose, eLife 2018, Kelso et al eLife 2017, Bao Genome Biol 2015), a reduction in BAF complex binding could be both a consequence and cause of reduced CEBPb binding, in which case the word 'direct' in line 269 may be an overstatement.

Minor Comments:

1. In Figure 3D and 3E and Figure 3 Supp 2E, CEBPb aggregate signal in the histogram doesn't match the heatmap. Why is this?

2. Motif enrichment analyses were done on the top 3000 significant peaks. Please clarify what significant means – using what statistical test?

Reviewer #2 (Remarks to the Author):

This paper utilizes genome-wide and mouse genetics studies to probe the functional interplay between two chromatin regulating complexes, Swi/Snf and Mll4 complex, using adipogenesis as a model system. Although their efforts to go deeper into the problem are praise-worthy, all main discoveries they claimed to have made represent only a minor advancement for the field. This may also explain their failure to cite and discuss all relevant prior reports fully. Both the actions of Sw/Snf to Mll4 function and the actions of Mll4-Utx to Swi/Snf function have been well characterized in numerous papers for the past decade. This paper suits better in a more specialized journal.

Reviewer #3 (Remarks to the Author):

This is an elegant study from Park and colleagues to demonstrate that chromatin remodeling complex BAF and methyltransferase MLL4 coordinate to regulate enhancer activation in adipocyte differentiation. The authors employed state-of-the-art epigenetic approaches in complementary adipocyte models to show that BAT and MLL4 reciprocally regulate each other's binding to the LDTF,

that is C/EBP β , enhancers to play essential role in regulating adipogenesis. Overall, this is a solid study with compelling and comprehensive evidences, although its physiological relevance is not established, and certain information/conclusions should be clarified.

1. In this study the authors studied adipogenesis but didn't distinguish brown vs. white adipocyte differentiation. These two processes overlap on the core adipogenic cascade but are quite different especially in the early lineage determination stage. Some adipocyte models in this manuscript used brown preadipocytes but many miss information. Are the findings specific to brown adipogenesis or extendable to white preadipocyte differentiation such as 3T3-L1, or to the upstream mesenchymal stem cell such as C3H10T1/2? It should be specified and discussed to make the conclusion accurate.

2. What's the physiological significance of this study? Are these epigenetic machinery/components changed in obesity?

3. Many of the figure legends are oversimplified without providing sufficient information. Specific adipocyte differentiation model information is missing in many places, so do statistical analysis information, n numbers etc.

4. The authors used different methods/models to ablate different factors, such as auxin-induced Tir1-dependent degradation for SMARCA4, PdgfRa-Cre for SMARCB1, Myf5-Cre for Pbrm1. The findings should be carefully integrated because of the variations among these KO approaches. For example, the author used Myf5-Cre to KO Pbrm1. Why not use PdgfRa-Cre to be consistent to Smarcb1 KO? Smarcb1;PdgfRa-Cre caused neonatal death whereas Pbrm1;Myf5-Cre KO showed minimal effect. Is it due to Cre difference?

5. Inf Fig. 2K, what happened to C/EBP β in Arid1a knockdown?

6. In Fig. 1B, Smarca4 is decreased during adipogenesis, but not in Fig. 2-Supple 1J. Please explain and clarify the adipocyte differentiation information.

7. In Fig. 3 and Fig. 4, what about the differentiation efficiency between control and KO? Look like impaired but please provide more evidence.

REVIEWER'S COMMENTS

Reviewer #1 (Remarks to the Author):

Park and colleagues test the functional interplay between the SWI/SNF complex and MLL4 during adipogenesis using several genetic model systems. They present evidence for the requirement of the canonical BAF complex (among the three SWI/SNF complex variants) in adipogenesis. This is demonstrated by a combination of transcriptomic profiling, red oil o staining, and in vivo characterization after acute deletion of *Smarca4* (using degron-based system), conditional deletion of *Smarca1*, and shRNA-mediated deletion of *Arid1a*. This requirement is not true of PBAF because conditional deletion of *Pbrm1* resulted in no or little defect during adipocyte differentiation. Using ChIP-seq studies, they observed binding of the BAF complex at active enhancers, some of which were pre-bound in pre-adipocytes and others were gained de novo during differentiation. Interestingly, when BAF complex binding sites are categorized based on MLL4 co-localization, the motifs enriched differ such that MLL+ sites are more enriched for adipogenic LDTFs and MLL- sites are more enriched for AP-1 family members. They then probed for the requirement of MLL4 or BAF subunit for reciprocal genomic binding. They report interdependence between binding of MLL4 and the BAF complex in adipocytes and in pre-adipocytes. Finally, they exogenously expressed CEBP β in pre-adipocytes to overcome effects of failed LDTF induction and assessed MLL4 and BAF complex binding, where they report reciprocal interdependence at CEBP β -gained enhancers. This is a nice study and thorough study looking at the interdependence of these factors with reciprocal genetic deletions.

We thank Reviewer #1 for careful reading of our manuscript and appreciate the highly constructive comments.

Major Comments:

1. Given that aspects of MLL3/4 activity are independent of their catalytic activity (Dorigi et al Mol Cell 2017), and that the maintenance of SMARCA4 binding was shown to be partially dependent on H3K4me in ESCs (Local Nat Genetics 2018), the authors should test whether SMARCA4/SS18 binding is reduced in MLL4 dCD reconstitution relative to MLL4 WT reconstitution in MLL3/4 dKO cells at gained active enhancers. This would be a nice addition to the model and test whether the mark itself is required for recruitment of BAF complexes in particular to de novo sites.

We agree that whether catalytic activity of MLL4 is required for BAF genomic binding is indeed an important question. However, reconstitution of WT or catalytic-dead mutants of MLL4 in KO cells is technically difficult due to its huge size (>20kb cDNA). The best way to address this question is to introduce knock-in mutations in the MLL3/4 SET domain to disrupt their catalytic activities. Even if we generate such mutant cells and test the BAF genomic binding, we still won't be able to conclude whether H3K4me1 modification itself is important or not, as we don't know whether there are non-histone substrates critical for MLL3/4 functions. We believe this fundamental question requires whole sets of new experiments to uncover catalytic activity-dependent and -independent roles of MLL4 in cell differentiation and enhancer regulation, and therefore, is beyond the scope of the current manuscript. We have a manuscript under review addressing the role of MLL3/4-mediated H3K4me1 on enhancer activation using embryonic stem cell differentiation as a model system

(<https://www.biorxiv.org/content/10.1101/2020.09.14.296558v1>).

2. Please include the sequencing depth and number of peaks called for all the ChIP-seq

experiments in the Reporting Summary. In addition, some of the heatmaps appear very washed out – please include a scale for all heatmaps.

We have included the sequencing depth and number of peaks for all ChIP-Seq experiments in the updated Reporting Summary. We have also included scale bars for heatmaps.

3. Replicate ChIP-seq datasets should be performed for ARID2 and BRD9 in Figure 1C. Overall, the number of peaks is quite a bit lower in D2 than D-3. It's unclear if this is due to biological or technical variation.

Following the reviewer's suggestion, we performed additional ChIP-Seq of ARID2 and BRD9. We confirmed that the number of BRD9 peaks is lower at D2 than at D-3. Consistent with our previous ChIP-Seq data in Figure 1 as well as findings in other studies, BRD9 binding sites identified from new experiments were enriched with CTCF motifs. We have included these new data in Supplementary Fig 1c-d. Unfortunately, we ran out of the old batch of ARID2 antibody (Santa Cruz, sc-166117X, lot J1415). We purchased a new batch (lot G2919) of the Santa Cruz antibody as well as an ARID2 antibody from Cell Signaling Technology (#82342). However, both failed in ChIP-Seq. It remains to be determined whether the reduced number of ARID2 binding peaks at D2 compared to D-3 is biologically relevant or due to technical variation.

4. It's difficult to assess quality of the mass spectrometry data by looking at hand-picked proteins. How many IP-MS replicates were done and was there an anti-IgG control used? The authors should use programs (SAINT, MaxQuant, etc.) to determine which proteins are statistically enriched in their IPs versus the IgG controls. Mass spectrometry data also needs to be deposited to a public repository like PRIDE (<https://www.ebi.ac.uk/pride/archive/>).

We would like to point out that the physical association between UTX, a subunit of MLL4 complex, and SMARCA4, a subunit of BAF complex, has been reported previously¹ and that we confirmed the results in this manuscript. Since IP-MS experiments were done seven years ago,

we were not able to deposit the data in a public database. Instead, we have provided the raw IP-MS data (list of identified proteins) in the Supplemental Data 1 and 2. We only did the IP-MS experiments once, and anti-IgG control was not used because it is difficult to prepare large quantities of ES cells for IP-MS. However, we confirmed the IP-MS data by independent IP-Western blot experiments (Fig. 4b, c).

5. In Figure 4 and 5, it's unclear whether the requirement for MLL4 is specific for MLL4⁺ sites. MLL4⁻ sites should be included in the heatmap analysis to establish the specific role of MLL4 in stabilizing BAF complex at sites where they are co-localized.

Following the reviewer's suggestion, we have included heatmaps for MLL4⁻ sites in the new Figures 5 and 6 (old Figures 4 and 5, respectively). In Figure 5, the requirement of MLL4 for BAF binding appears to be specific for MLL4⁺ adipogenic enhancers. These sets of enhancers are mostly *de novo* active enhancers emerging after inducing adipogenesis. Therefore, this data suggests that MLL4 is specifically required for *de novo* binding of BAF during adipogenesis. However, in Figure 6, both MLL4⁺ and MLL4⁻ active enhancers in preadipocytes show decreased BAF binding in *Mll4* KO cells, suggesting that deletion of *Mll4* has both primary and secondary effects on maintaining BAF binding on active enhancers in preadipocytes. We have modified the manuscript accordingly.

6. In Figure 6D, CEBPb binding appears to be reduced in the CEBPb OE/*Mll4*-deleted samples compared to CEBPb OE/no cre treatment, despite the fact that the aggregate signal overlays on top of each other (Figure 6E). Please include box plots of log₂fc as done in other figures. Given several previous studies showing the interdependence of TF and BAF complex binding (King and Klose, eLife 2018, Kelso et al eLife 2017, Bao Genome Biol 2015), a reduction in BAF

complex binding could be both a consequence and cause of reduced CEBPb binding, in which case the word 'direct' in line 269 may be an overstatement.

We have included box plots of log2fc in the new Figure 7f (old Figure 6f). While SMARCA4 and ARID1A binding obviously decreased in *Mll4* KO cells, C/EBPβ binding only mildly decreased, which could be due to reduced BAF binding in the *Mll4* KO cells. As the reviewer pointed out, a reduction in BAF binding in *Mll4* KO cells could be both a consequence and cause of reduced C/EBPβ binding. We have taken out the word 'direct' in the Results section and also modified the Discussion section with additional references.

Minor Comments:

1. In Figure 3D and 3E and Figure 3 Supp 2E, CEBPb aggregate signal in the histogram doesn't match the heatmap. Why is this?

On BAF-prebound active enhancers (Figure 3d-e), C/EBPβ occupies much fewer numbers of regions at D-3 (1,309) than at D2 (3,509), but with higher intensities. This becomes evident if we draw heat maps according to C/EBPβ intensity at D-3 (Figure 1 for reviewer).

Figure 1 for reviewer

SMARCA4⁺SS18⁺ AE (D2)

2. Motif enrichment analyses were done on the top 3000 significant peaks. Please clarify what significant means – using what statistical test?

We used the top 3000 significant peaks based on the FDR value provided by SICER². Briefly, FDR was calculated using p-value adjusted for multiple testing, and p-value was based on a test using the Poisson distribution. We have added this information in Materials and Methods under the Motif analysis section.

Reviewer #2 (Remarks to the Author)

This paper utilizes genome-wide and mouse genetics studies to probe the functional interplay between two chromatin regulating complexes, Swi/Snf and Mll4 complex, using adipogenesis as a model system. Although their efforts to go deeper into the problem are praise-worthy, all main discoveries they claimed to have made represent only a minor advancement for the field. This may also explain their failure to cite and discuss all relevant prior reports fully. Both the actions of Sw/Snf to Mll4 function and the actions of Mll4-Utx to Swi/Snf function have been well characterized in numerous papers for the past decade. This paper suits better in a more specialized journal.

We greatly appreciate Reviewer #2's efforts on evaluating our manuscript.

Several papers have described the relationship between SWI/SNF and MLL4 complexes. It was reported that H3K4me1-marked nucleosome can pull down SWI/SNF components including SMARCA4 from HeLa nuclear extract and that disruption of catalytic activities of MLL3/4 reduces SMARCA4 binding on H3K4me1⁺ distal regions in undifferentiated mouse embryonic stem cells³. On the other hand, it was also shown that BAF (SMARCA4 and SS18) does not require pre-marked H3K4me1 for binding and that targeting of ectopic SMARCA4 in SMARCA4-deficient cancer cells establishes MLL3/4 genomic binding and active enhancer marks *de novo*⁴. Another study showed that UTX physically associates with SMARCA4 in mouse myeloid cells and that loss of UTX leads to changes in chromatin accessibility and enhancer activation¹.

In this manuscript, using adipogenesis and the pioneer factor C/EBP β -mediated enhancer activation as model systems, we establish an interdependent relationship between BAF and

MLL4 in promoting enhancer activation through lineage-determining transcription factors. Our findings not only rectify seemingly conflicting results from previous separate studies but also present an important conceptual advance in the understanding of enhancer activation during cell differentiation. In addition, we report distinct genomic localizations of SWI/SNF complexes during differentiation using adipogenesis as a model. Further, we report for the first time that BAF, but not PBAF, is required for adipogenesis *in vivo*, which indicates distinct functions of SWI/SNF complexes in cell differentiation and development. We believe these new findings are both novel and significant enough to advance current knowledge on the roles of MLL3/4 and SWI/SNF in enhancer regulation and cell differentiation.

Reviewer #3 (Remarks to the Author):

This is an elegant study from Park and colleagues to demonstrate that chromatin remodeling complex BAF and methyltransferase MLL4 coordinate to regulate enhancer activation in adipocyte differentiation. The authors employed state-of-the-art epigenetic approaches in complementary adipocyte models to show that BAT and MLL4 reciprocally regulate each other's binding to the LDTF, that is C/EBP β , enhancers to play essential role in regulating adipogenesis. Overall, this is a solid study with compelling and comprehensive evidences, although its physiological relevance is not established, and certain information/conclusions should be clarified.

We thank Reviewer #3 for careful reading of our manuscript and appreciate the highly constructive comments.

1. In this study the authors studied adipogenesis but didn't distinguish brown vs. white adipocyte

differentiation. These two processes overlap on the core adipogenic cascade but are quite different especially in the early lineage determination stage. Some adipocyte models in this manuscript used brown preadipocytes but many miss information. Are the findings specific to brown adipogenesis or extendable to white preadipocyte differentiation such as 3T3-L1, or to the upstream mesenchymal stem cell such as C3H10T1/2? It should be specified and discussed to make the conclusion accurate.

We have updated cell line information where brown preadipocytes were used in figure legends. We used immortalized brown preadipocytes as model systems. While these cells express brown adipocyte-specific genes such as *Ucp1* after differentiation, the expression level is extremely low compared to that in BAT *in vivo* (~1/5000)⁵. In addition, immortalized brown preadipocytes and 3T3-L1 white preadipocytes share common transcriptional features during adipogenesis. Expression patterns of common adipogenic markers such as *Pparg*, *Cebpa*, and *Fabp4* and the enhancer activation dynamics are similar in immortalized brown preadipocytes and 3T3-L1 white preadipocytes. Our study focuses on general adipogenesis rather than brown-specific gene induction. We showed that BAF, but not PBAF, is required for adipogenesis of preadipocytes *in vitro* and in mice. It remains to be determined whether this is the case in upstream mesenchymal stem cells such as C3H10T1/2.

2. What's the physiological significance of this study? Are these epigenetic machinery/components changed in obesity?

We searched the GEO database to find out whether expressions of BAF and MLL4 components change in obesity. A RNA-Seq dataset from 6 lean and 6 obese human subjects (GSE133099) showed that only *BCL7A* and *SMARCD1* expression was significantly increased and decreased, respectively, in adipocytes from obese human subjects (Table 1 for reviewer). In the same dataset, the expression of master adipogenic TFs *PPARG* and *CEBPA* did not change significantly. Another RNA-Seq dataset from brown adipose tissues of 3 lean and 3 high fat diet-

induced obese mice (GSE112740) also showed no significant changes in BAF or MLL4 subunits expression (Table 2 for reviewer)⁶.

A number of studies have shown that BAF and MLL4 subunits are frequently mutated in human cancers^{7,8}, suggesting that enhancer function is critical for maintaining normal physiology. The physiological roles of BAF and MLL4 components and the interplay between BAF and MLL4 in human adipose tissue development and obesity remain an open question.

Table 1 for reviewer. BAF and MLL4 components expression in obese vs lean human adipocytes

	gene id	gene symbol	log2(obese/lean)	p-value
BAF	ENSG00000075624	ACTB	-0.29	0.12
	ENSG00000136518	ACTL6A	-0.17	0.30
	ENSG00000117713	ARID1A	-0.02	0.94
	ENSG00000049618	ARID1B	0.28	0.11
	ENSG00000110987	BCL7A	0.98	0.00
	ENSG00000106635	BCL7B	0.02	0.84
	ENSG00000099385	BCL7C	0.14	0.49
	ENSG00000011332	DPF1	1.49	0.73
	ENSG00000133884	DPF2	-0.07	0.52
	ENSG00000205683	DPF3	0.04	0.94

	ENSG00000080503	SMARCA2	-0.01	0.93
	ENSG00000127616	SMARCA4	0.17	0.28
	ENSG00000153147	SMARCA5	-0.10	0.45
	ENSG00000099956	SMARCB1	0.05	0.67
	ENSG00000173473	SMARCC1	0.07	0.60
	ENSG00000139613	SMARCC2	0.30	0.12
	ENSG00000066117	SMARCD1	-0.41	0.01
	ENSG00000108604	SMARCD2	0.14	0.41
	ENSG00000073584	SMARCE1	-0.07	0.50
	ENSG00000141380	SS18	0.00	0.98
MLL4	ENSG00000129691	ASH2L	-0.23	0.19
	ENSG00000162961	DPY30	0.10	0.58
	ENSG00000147050	KDM6A	0.01	0.97
	ENSG00000167548	KMT2D	-0.16	0.46
	ENSG00000198646	NCOA6	0.03	0.91
	ENSG00000280789	PAGR1	0.32	0.55
	ENSG00000157212	PAXIP1	-0.19	0.17
	ENSG00000117222	RBBP5	0.00	0.98
	ENSG00000196363	WDR5	-0.02	0.89

Adipogenic TFs	ENSG00000132170	PPARG	-0.37	0.36
	ENSG00000245848	CEBPA	0.36	0.45

Table 2 for reviewer. BAF and MLL4 components expression in obese vs lean mouse

BAT

	gene id	gene symbol	log2(obese/lean)	p-value
BAF	ENSMUSG00000029580	Actb	0.71	0.33
	ENSMUSG00000027671	Actl6a	0.42	0.30
	ENSMUSG00000007880	Arid1a	-0.20	0.67
	ENSMUSG00000069729	Arid1b	-0.05	0.84
	ENSMUSG00000029438	Bcl7a	-0.06	0.87
	ENSMUSG00000029681	Bcl7b	0.09	0.69
	ENSMUSG00000030814	Bcl7c	-0.64	0.44
	ENSMUSG00000030584	Dpf1	-0.41	0.75
	ENSMUSG00000024826	Dpf2	0.22	0.54
	ENSMUSG00000021221	Dpf3	-0.20	0.79
	ENSMUSG00000024921	Smarca2	-0.82	0.06
	ENSMUSG00000032187	Smarca4	0.06	0.86
	ENSMUSG00000031715	Smarca5	0.19	0.66

	ENSMUSG00000000902	Smarcb1	0.07	0.83
	ENSMUSG00000032481	Smarcc1	-0.23	0.63
	ENSMUSG00000025369	Smarcc2	-0.22	0.05
	ENSMUSG00000023018	Smarcd1	0.10	0.74
	ENSMUSG00000078619	Smarcd2	-0.41	0.07
	ENSMUSG00000037935	Smarce1	0.21	0.66
	ENSMUSG00000037013	Ss18	0.04	0.85
MLL4	ENSMUSG00000031575	Ash2l	-0.22	0.67
	ENSMUSG00000037369	Kdm6a	0.17	0.68
	ENSMUSG00000048154	kmt2d	-0.47	0.38
	ENSMUSG00000038369	Ncoa6	-0.28	0.51
	ENSMUSG00000030680	Pagr1a	0.75	0.31
	ENSMUSG00000002221	Paxip1	0.08	0.83
	ENSMUSG00000024067	Dpy30	-0.05	0.82
	ENSMUSG00000026439	Rbbp5	0.26	0.53
	ENSMUSG00000026917	Wdr5	0.15	0.71
Adipogenic TFs	ENSMUSG00000000440	Pparg	0.20	0.68
	ENSMUSG00000034957	Cebpa	-0.90	0.24

3. Many of the figure legends are oversimplified without providing sufficient information. Specific adipocyte differentiation model information is missing in many places, so do statistical analysis information, n numbers etc.

We have updated original bar graphs with bar/plot graphs for Fig. 2k, Supplementary Fig. 2f, and Supplementary Fig. 3c, e and f and indicated n numbers in the revised manuscript. We also added information on specific adipogenesis models and statistical analysis in Figure legends.

4. The authors used different methods/models to ablate different factors, such as auxin-induced Tir1-dependent degradation for SMARCA4, PdgfRa-Cre for SMARCB1, Myf5-Cre for Pbrm1.

The findings should be carefully integrated because of the variations among these KO approaches. For example, the author used Myf5-Cre to KO Pbrm1. Why not use PdgfRa-Cre to be consistent to Smarcb1 KO? Smarcb1;PdgfRa-Cre caused neonatal death whereas Pbrm1;Myf5-Cre KO showed minimal effect. Is it due to Cre difference?

It has been shown that over 90% of mature brown adipocytes arise from Myf5-Cre expressing precursors in mice^{9,10}. Myf5-Cre is the best tool to test whether SMARCB1 is required for brown adipose tissue development in mice. However, since the *Smarcb1* gene is located on chromosome 10, the same chromosome as the *Myf5-Cre* transgene, there is a technical problem of low recombination frequency (1/24, according to the distance between the two loci). Therefore, we instead used *PdgfRa-Cre* to delete *Smarcb1*, as this transgene is expressed in brown adipocyte precursors¹¹. However, *PdgfRa-Cre* is also expressed in several other tissues during embryogenesis, making it less desirable due to potential off-target phenotypes from other cell types including cranial and cardiac neural crest cells¹². Neonatal death of *Smarcb1*^{fl/fl}; *PdgfRa-Cre* mice is caused by *Smarcb1* deletion in cranial neural crest cells, which led to defects in coronal sutures, but not by *Smarcb1* deletion in brown adipose precursors. Further, we also confirmed the *in vivo* phenotypes by performing adipogenesis using

preadipocytes isolated from BAT in cell culture. We have updated the Discussion section accordingly.

5. In Fig. 2K, what happened to C/EBP β in *Arid1a* knockdown?

We have performed qRT-PCR to check *Cebpb* expression during adipogenesis (D0, 2h, 4h, 24h, D2, D7). Knockdown of *Arid1a* doesn't affect *Cebpb* expression during adipogenesis. We have included this new data in Fig. 2k and updated the manuscript accordingly.

6. In Fig. 1B, *Smarca4* is decreased during adipogenesis, but not in Fig. 2-Supple 1J. Please explain and clarify the adipocyte differentiation information.

We checked additional RNA-Seq data sets from 3T3-L1 (GSE87113)¹³ and a different brown preadipocyte cell line (GSE99101)¹⁴. In these two data sets and the data set originally included, *Smarca4* mRNA levels decreased about 2-fold at D7 compared to those at D-3 (Fig 1b, new Supplementary Fig 1a). There is no obvious decrease in *Smarca4* levels at D2 of adipogenesis. Supplementary Fig 2j (old Fig 2-Supple 1J) presents RNA-Seq data collected at D-3 and D2 of adipogenesis in *Smarcb1*^{fl/fl} cells and shows no decrease in *Smarca4* levels at D2 of adipogenesis, which is consistent with data in Fig 1b and Supplementary Fig 1a.

7. In Fig. 3 and Fig. 4, what about the differentiation efficiency between control and KO? Look like impaired but please provide more evidence.

We have included Oil Red O staining data in Supplementary Fig 6. Consistent with our previous data, knockout of *Mll3/Mll4* blocked adipogenesis¹⁵. We have updated the manuscript accordingly.

References

- 1 Gozdecka, M. *et al.* UTX-mediated enhancer and chromatin remodeling suppresses myeloid leukemogenesis through noncatalytic inverse regulation of ETS and GATA programs. *Nat Genet* **50**, 883-894, doi:10.1038/s41588-018-0114-z (2018).
- 2 Zang, C. *et al.* A clustering approach for identification of enriched domains from histone modification ChIP-Seq data. *Bioinformatics* **25**, 1952-1958, doi:10.1093/bioinformatics/btp340 (2009).
- 3 Local, A. *et al.* Identification of H3K4me1-associated proteins at mammalian enhancers. *Nat Genet* **50**, 73-82, doi:10.1038/s41588-017-0015-6 (2018).
- 4 Pan, J. *et al.* The ATPase module of mammalian SWI/SNF family complexes mediates subcomplex identity and catalytic activity-independent genomic targeting. *Nat Genet* **51**, 618-626, doi:10.1038/s41588-019-0363-5 (2019).
- 5 Lai, B. *et al.* MLL3/MLL4 are required for CBP/p300 binding on enhancers and super-enhancer formation in brown adipogenesis. *Nucleic Acids Res* **45**, 6388-6403, doi:10.1093/nar/gkx234 (2017).
- 6 Cao, J. *et al.* Global Transcriptome Analysis of Brown Adipose Tissue of Diet-Induced Obese Mice. *Int J Mol Sci* **19**, doi:10.3390/ijms19041095 (2018).
- 7 Kadoch, C. *et al.* Proteomic and bioinformatic analysis of mammalian SWI/SNF complexes identifies extensive roles in human malignancy. *Nat Genet* **45**, 592-601, doi:10.1038/ng.2628 (2013).
- 8 Wilson, B. G. & Roberts, C. W. SWI/SNF nucleosome remodellers and cancer. *Nat Rev Cancer* **11**, 481-492, doi:10.1038/nrc3068 (2011).
- 9 Seale, P. *et al.* PRDM16 controls a brown fat/skeletal muscle switch. *Nature* **454**, 961-967, doi:10.1038/nature07182 (2008).
- 10 Sanchez-Gurmaches, J. *et al.* PTEN loss in the Myf5 lineage redistributes body fat and reveals subsets of white adipocytes that arise from Myf5 precursors. *Cell Metab* **16**, 348-362, doi:10.1016/j.cmet.2012.08.003 (2012).
- 11 Berry, R. & Rodeheffer, M. S. Characterization of the adipocyte cellular lineage in vivo. *Nat Cell Biol* **15**, 302-308, doi:10.1038/ncb2696 (2013).
- 12 Tallquist, M. D. & Soriano, P. Cell autonomous requirement for PDGFRalpha in populations of cranial and cardiac neural crest cells. *Development* **130**, 507-518, doi:10.1242/dev.00241 (2003).
- 13 Park, Y. K. *et al.* Distinct Roles of Transcription Factors KLF4, Krox20, and Peroxisome Proliferator-Activated Receptor gamma in Adipogenesis. *Mol Cell Biol* **37**, doi:10.1128/MCB.00554-16 (2017).
- 14 Lee, J. E. *et al.* Brd4 binds to active enhancers to control cell identity gene induction in adipogenesis and myogenesis. *Nat Commun* **8**, 2217, doi:10.1038/s41467-017-02403-5 (2017).
- 15 Lee, J. E. *et al.* H3K4 mono- and di-methyltransferase MLL4 is required for enhancer activation during cell differentiation. *Elife* **2**, e01503, doi:10.7554/eLife.01503 (2013).

REVIEWERS' COMMENTS

Reviewer #1 (Remarks to the Author):

The authors have addressed my comments. I commend them on this nice work.

Reviewer #3 (Remarks to the Author):

The authors have done an excellent job to address my comments.